# Global Policy-Space Response Oracles for Two-Player Zero-Sum Games

**Junyu Zhang** [1]  **Feihong Yang** [2]  **Jian Wang** [1]  **Chao Wang** [2]  **Xudong Zhang** [1]

## Abstract

The Policy-Space Response Oracles (PSRO) framework scales equilibrium computation to large zero-sum games by iteratively expanding a restricted strategy set using deep reinforcement learning (DRL). A central challenge is to construct, under limited computational budgets, a small strategy population whose induced game well approximates the full game. Existing PSRO variants typically expand the population using best responses to meta-strategies computed from restricted-game payoffs, which can lead to inefficient expansions that provide limited global improvement. We propose to guide population expansion by directly evaluating the post-expansion population quality. Specifically, we adopt Population Exploitability (PE) to measure how well a restricted strategy set represents the full game, and introduce a two-phase exploration–selection framework that explicitly minimizes PE during expansion. We instantiate this framework as Global PSRO, a practical DRL-based algorithm that efficiently generates candidate responses and estimates PE via parameter-sharing conditional neural networks. Experiments across multiple two-player zero-sum games show that Global PSRO achieves lower exploitability and approximates Nash equilibria with significantly fewer policy iterations than prior PSRO methods. The experimental code is available at https://github.com/Zhangjy1997/GlobalPSRO.

## 1. Introduction

Computing equilibria in large-scale games is challenging due to enormous strategy spaces and the intractability of exhaustive analysis (Von Stengel, 2002; Avis et al., 2010). The Policy-Space Response Oracle (PSRO) framework improves scalability by iteratively solving restricted games over a growing strategy population, where best responses (BR) are often learned via deep reinforcement learning (DRL) (Lanctot et al., 2017).

A central difficulty in PSRO lies in *strategy population construction*: how to assemble, under limited computational budgets, a small set of strategies whose induced restricted game faithfully represents the strategic structure of the full game. In standard PSRO, each iteration adds a BR to a meta-strategy computed by a meta-strategy solver (MSS), typically the Nash equilibrium (NE) of the current restricted game. Since the restricted game is only a local approximation, a BR to its equilibrium can be locally strong yet contribute little to improving the approximation quality of the full game, leading to inefficient expansion. Many PSRO variants modify the MSS to encourage exploration or reduce overfitting (Lanctot et al., 2017; Wang et al., 2019; Wang & Wellman, 2023; Wright et al., 2019; Li et al., 2023), but they still rely primarily on restricted-game payoffs and do not explicitly evaluate whether an expansion improves full-game approximation; in the worst case, such restricted-game-based expansions may require adding essentially the entire strategy space to converge (Sec. 3.1). Methods that incorporate extra global information, such as Anytime PSRO (McAleer et al., 2022), aim to find meta-strategies that are hard for full-game opponents to exploit, yet still expand by responding to a single current meta-strategy without directly assessing the population after expansion.

We instead make expansion decisions by explicitly optimizing population quality *after* adding a candidate strategy. This is directly aligned with the primary goal of PSRO-style methods, namely constructing a small population whose induced game approximates a full game equilibrium. We measure population quality with *Population Exploitability* (PE), the minimum exploitability achievable by mixtures supported on the population against full-game opponents. This leads to a two-phase *exploration–selection* framework. In the exploration phase, we generate a batch of candidate responses by training against multiple meta-strategies given by different kinds of MSSs. In the selection phase, we evaluate the PE of each *post-expansion* population obtained by hypothetically adding a candidate strategy, and select the candidate that minimizes this post-expansion PE. This directly aligns the expansion with improving population-level

[1]Department of Electronic Engineering, Tsinghua University, Beijing, China [2]Qiyuan Lab, Beijing, China. Correspondence to: Chao Wang <wangchao1@qiyuanlab.com>.

*Proceedings of the 43rd International Conference on Machine Learning*, Seoul, South Korea. PMLR 306, 2026. Copyright 2026 by the author(s).

equilibrium approximation. Furthermore, the PE estimation procedure naturally yields a BR to the selected population mixture in the full game. We also add the BR to the population, which corresponds to executing one standard Anytime-PSRO expansion alongside the selected candidate. Overall, two strategies are obtained in the selection phase to expand the restricted game.

We instantiate this framework as **Global PSRO**, a practical DRL-based algorithm for large-scale two-player zero-sum games. To make multi-candidate training and PE-based selection feasible, we use parameter sharing with conditional policy models for candidate generation and PE estimation, and adopt a regularized PE-based score to make selection robust to PE estimation errors. Experiments across several benchmark games show that Global PSRO reaches substantially better equilibrium approximations than existing PSRO variants under matched interaction budgets.

Our contributions can be summarized as follows:

- We analyze existing PSRO meta-strategy solvers from the perspective of global information utilization, and show that relying solely on restricted-game information can lead to highly inefficient population expansion.

- We formulate strategy population construction as an explicit optimization problem that is directly aligned with the primary objective of PSRO, namely minimizing the PE of the *expanded* population after each addition, and propose an exploration–selection framework that ranks candidates by their estimated post-expansion PE.

- We introduce Global PSRO, a scalable DRL-based instantiation with parameter-shared candidate generation and regularized PE estimation for practical multi-candidate selection, and empirically demonstrate superior iteration and sample efficiency across a range of two-player zero-sum games.

## 2. Preliminaries

### 2.1. Games

We consider finite $n$-player normal-form games. A game is defined by a tuple $\mathcal{G} = (\mathcal{N}, \{\Pi_i\}_{i=1}^n, \{U_i\}_{i=1}^n)$, where $\mathcal{N} = \{1, \dots, n\}$ denotes the set of players, $\Pi_i$ is the pure strategy set of player $i$, and $U_i : \Pi_1 \times \cdots \times \Pi_n \to \mathbb{R}$ is the payoff function. We focus on two-player zero-sum games, where $U_1 = -U_2$. A two-player game is symmetric if $\Pi_1 = \Pi_2 = \Pi$ and the payoff functions are identical up to player exchange, i.e., $U_1(\pi_i, \pi_j) = U_2(\pi_j, \pi_i)$ for all $\pi_i, \pi_j \in \Pi$.

A mixed strategy $\sigma_i \in \Delta(\Pi_i)$ is a probability distribution over $\Pi_i$, and $\sigma = (\sigma_1, \sigma_2)$ denotes a joint mixed strategy. For $\pi_i \in \Pi_i$, we write $\sigma_i(\pi_i)$ for its probability and

$\text{supp}(\sigma_i) = \{\pi_i \in \Pi_i : \sigma_i(\pi_i) > 0\}$ for the support. The expected payoff to player $i$ under $\sigma$ is denoted by $U_i(\sigma)$. Given an opponent strategy $\sigma_{-i}$, a (pure) BR of player $i$ is defined as

$$\mathcal{B}_i(\sigma_{-i}) = \arg \max_{\pi_i \in \Pi_i} U_i(\pi_i, \sigma_{-i}). \tag{1}$$

### 2.2. Restricted Games and Meta-Games

PSRO operates by maintaining, for each player $i$, a restricted strategy set $\Pi_i^r \subseteq \Pi_i$. The Cartesian product $\Pi^r = \Pi_1^r \times \Pi_2^r$ induces a *restricted game*, whose payoff matrix is defined by evaluating $U_i$ on strategy profiles in $\Pi^r$.

Solving the restricted game yields a *meta-strategy* $\sigma \in \Delta(\Pi^r)$, typically computed by a MSS such as a NE solver. This restricted game, together with its meta-strategy, is often referred to as the *meta-game*. Importantly, the meta-game is only an approximation of the full game, and its quality depends entirely on the choice of the restricted strategy set $\Pi^r$.

In each iteration of PSRO, new strategies are generated by computing (approximate) BRs to the current meta-strategy, commonly using DRL. These strategies are then added to $\Pi^r$, and the meta-game is recomputed.

### 2.3. Exploitability and Population Exploitability

The exploitability of a joint mixed strategy $\sigma$ is defined as

$$\epsilon(\sigma) = \frac{1}{|\mathcal{N}|} \sum_{i \in \mathcal{N}} \max_{\pi_i \in \Pi_i} \left[ U_i(\pi_i, \sigma_{-i}) - U_i(\sigma) \right]. \tag{2}$$

A strategy $\sigma^\star$ is a NE if and only if $\epsilon(\sigma^\star) = 0$.

Exploitability is a property of an individual mixed strategy. In contrast, the primary goal of PSRO is to construct *restricted strategy populations* that approximate the full game well. To measure the approximation quality of a restricted strategy set $\Pi^r$ as an approximation of the full game, a commonly used population-level quantity is PE, defined as

$$\mathcal{PE}(\Pi^r; \mathcal{G}) = \min_{\sigma \in \Delta(\Pi^r)} \epsilon(\sigma). \tag{3}$$

PE measures the smallest exploitability attainable by any mixed strategy whose support is restricted to $\Pi^r$, and thus quantifies the approximation quality of the meta-game induced by the strategy population.

## 3. Revisiting Existing PSRO

The effectiveness of PSRO critically depends on how new strategies are selected and added to the restricted population. This choice is governed by the MSS, which determines the mixed strategy to which response oracles are trained.

In this section, we revisit existing PSRO variants through a unifying lens: *how much payoff information from the full game is used when selecting population expansions*. This perspective reveals fundamental limitations of prior approaches and motivates the need for explicitly evaluating population-level improvements.

### 3.1. Restricted-Game-Based PSRO

Many PSRO instantiations compute the meta-strategy using only the payoff matrix of the current restricted game. Representative MSSs include restricted-game NE (McMahan et al., 2003), projected replicator dynamics (PRD) (Lanctot et al., 2017), AlphaRank (Muller et al., 2019), and Uniform (Heinrich & Silver, 2016). We refer to this class as *Restricted-Game-Based MSSs (RGB-MSSs)*.

Formally, an RGB-MSS maps the restricted-game payoff matrix to a mixed strategy, without accessing payoff information involving strategies outside the current population.

**Definition 3.1.** In a two-player symmetric zero-sum game, an RGB-MSS $\mathcal{M}$ is a family of functions

$$\mathcal{M} = \{f_d \mid f_d : \mathbb{R}^{d \times d} \to \Delta^{d-1}\}_{d=1}^{\infty}, \qquad (4)$$

where $\Delta^{d-1} = \{\mathbf{x} \in \mathbb{R}^d \mid \mathbf{x} \geq 0, \sum_i x_i = 1\}$ denotes the probability simplex.

To study the effect of the MSS under an idealized oracle, we assume access to an exact BRO. We denote the resulting PSRO procedure by $\mathcal{D}(\mathcal{M}, \mathcal{G}, \pi_0)$, where $\mathcal{M}$ is the MSS, $\mathcal{G}$ the underlying game, and $\pi_0$ the initial strategy. PSRO is said to achieve a NE when the PE of the restricted strategy set reaches zero.

Although RGB-MSSs are computationally appealing, their reliance on restricted-game payoffs alone can fundamentally limit progress toward a full-game equilibrium. Intuitively, since strategies outside the current population are ignored when computing the meta-strategy, the resulting BRs may appear effective locally while contributing little to reducing global exploitability. The following result formalizes this limitation.

**Theorem 3.2** (Worst-case behavior in the PSRO with RGB-MSS). *Given any RGB-MSS $\mathcal{M}$, initial strategy $\pi_0$, and $N \in \mathbb{N}$, there exists an $N \times N$ two-player symmetric zero-sum game $\mathcal{G}$ such that at least one of the following holds: (i) the PSRO procedure $\mathcal{D}(\mathcal{M}, \mathcal{G}, \pi_0)$ fails to converge to a NE; or (ii) PSRO reaches a NE only after incorporating all pure strategies into the restricted population, i.e., after at least $N-1$ iterations.*

The proof is provided in Appendix A.1. This result shows that inefficient population expansion is not an artifact of a particular algorithm, but a structural limitation shared by all RGB-MSSs.

Importantly, such worst-case behavior is not inherent to the games themselves. For some instances where an RGB-MSS exhibits arbitrarily slow progress, an alternative MSS can reach equilibrium in only a few iterations. We have the following theorem.

**Theorem 3.3.** *Fix $N \in \mathbb{N}$, an odd integer $S$[1] with $1 \leq S \leq N$, an RGB-MSS $\mathcal{M}$, and an initial strategy $\pi_0$. There exists an $N \times N$ two-player symmetric zero-sum game $G$ that admits a NE $(\sigma^\star, \sigma^\star)$ with $|\text{supp}(\sigma^\star)| = S$, on which $\mathcal{D}(\mathcal{M}, G, \pi_0)$ exhibits the worst-case behavior in Theorem 3.2; yet there exists an alternative MSS $\mathcal{M}'$ such that $\mathcal{D}(\mathcal{M}', G, \pi_0)$ reaches a NE within $\min\{S+2, N-1\}$ iterations.*

The proof is given in Appendix A.2. Figure 1 shows adversarially constructed games with different equilibrium support sizes: each instance is designed to slow down the MSS named in the subcaption, while the instance-specific solver $\mathcal{M}'$ (from Theorem 3.3) reaches equilibrium within the corresponding iteration bound.

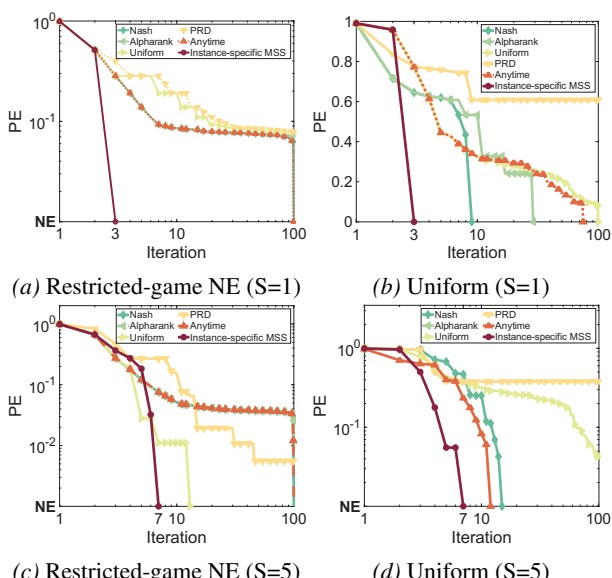

*(a) Restricted-game NE (S=1)*  *(b) Uniform (S=1)*

*(c) Restricted-game NE (S=5)*  *(d) Uniform (S=5)*

*Figure 1.* PE trajectories of PSRO under various MSSs on adversarially constructed $100 \times 100$ zero-sum games. Each subfigure is tailored to the MSS in its subcaption. See Appendix B.1 for more results.

Notably, the instance-specific MSS $\mathcal{M}'$ in Theorem 3.3 requires access to full-game payoff information and is therefore impractical in large-scale settings. Nevertheless, it highlights a key insight: *incorporating appropriate global information into population expansion can dramatically*

---

[1]In the generic non-degenerate case, symmetric zero-sum games admit only equilibrium supports of odd size (Brandl, 2017); therefore we consider odd $S$ here. For general two-player zero-sum games, the theorem can be extended to any positive support size.

*improve iteration efficiency.*

## 3.2. Semi-Restricted-Game-Based PSRO

Several PSRO variants incorporate partial full-game information when computing the meta-strategy. A useful abstraction is the semi-restricted setting, where the meta-strategy is supported on the restricted population, but exploitability is evaluated against full-game opponents. We refer to such methods as *Semi-Restricted-Game-Based MSSs (SRGB-MSSs).*

This category includes methods such as Anytime PSRO (McAleer et al., 2022) and EPSRO (Zhou et al., 2022), which aim to find a mixed strategy over $\Pi^r$ that minimizes exploitability against the full strategy space:

$$\sigma^* = \arg \min_{\sigma \in \Delta(\Pi^r)} \max_{\pi \in \Pi} U(\pi, \sigma). \qquad (5)$$

While SRGB-MSSs explicitly incorporate full-game opponents when selecting the meta-strategy, they still follow a fixed rule for choosing $\sigma$ and generate a response solely to that strategy. Crucially, these methods do not explicitly evaluate whether adding the resulting BR meaningfully improves the quality of the *strategy population* itself. As PSRO fundamentally aims to construct an effective population with minimal global exploitability, this gap motivates a more direct, population-level evaluation of candidate expansions.

## 4. Methods

Motivated by the limitations of existing PSRO variants identified in Section 3, we focus on *population expansion*: which strategy should be added next so that the *expanded* population best approximates the full-game equilibrium. We formalize this goal by selecting candidates that minimize the post-expansion PE. This leads to an optimization objective and an exploration–selection framework, which we instantiate as **Global PSRO**.

## 4.1. Optimization Objective

We aim to build a restricted strategy set that approximates the full-game equilibrium. We evaluate it using PE and minimize PE after each expansion step.

Formally, let $\Pi^r$ be the current restricted strategy set. We select the next strategy as

$$\pi^* \in \arg \min_{\pi \in \mathcal{B}(\Delta(\Pi^r))} \mathcal{PE}(\Pi^r \cup \{\pi\}; \mathcal{G}), \qquad (6)$$

where $\mathcal{B}(\Delta(\Pi^r))$ denotes the set of BRs induced by mixtures over $\Delta(\Pi^r)$.

Eq. (6) differs from standard PSRO updates in that candidates are ranked by the *population-level* improvement they

enable after being added, rather than by how well they respond to a single meta-strategy. This directly aligns strategy exploration with the objective of improving full-game equilibrium approximation.

## 4.2. Framework

Directly optimizing Eq. 6 is intractable in large-scale games. We alternatively approximate it by a two-phase exploration-selection framework that iteratively expands a strategy population (Fig. 2).

### 4.2.1. PHASE I: EXPLORATION

The exploration phase aims to generate a diverse set of candidate expansions for the current restricted strategy set $\Pi^r$. We construct a pool of meta-strategies

$$\mathcal{S}_t = \{\sigma_k\}_{k=1}^K \subset \Delta(\Pi^r), \qquad (7)$$

that balances (i) *exploitation* of the current restricted game via principled meta-strategies and (ii) *exploration* to discover diverse BRs beyond those obtained from a single selection rule. For each $\sigma_k \in \mathcal{S}_t$, we compute a BR $\pi_k \in \mathcal{B}(\sigma_k)$ and form the candidate expansion

$$\Pi_k^+ = \Pi^r \cup \{\pi_k\}. \qquad (8)$$

The exploration phase aims to diversify the meta-strategies in $\mathcal{S}_t$ as much as possible. It is useful to understand how BRs vary over the simplex $\Delta(\Pi^r)$ in finite games. A key property is local stability: if a mixture admits a unique BR, that response remains optimal under small perturbations. As a result, $\Delta(\Pi^r)$ decomposes into regions on which the BR is constant, so a finite pool $\mathcal{S}_t$ can cover large portions of the simplex without an exhaustive sweep.

**Proposition 4.1.** *Consider a simplex $\Delta(\Pi^r)$. If a mixed strategy $\sigma \in \Delta(\Pi^r)$ admits a unique BR $\pi^\star$, then there exists $\varepsilon > 0$ such that $\pi^\star$ remains the unique BR for all mixed strategies in $\{\sigma' \in \Delta(\Pi^r) \mid \|\sigma' - \sigma\|_2 < \varepsilon\}$.*

The proof is deferred to Appendix A.3. Figure 3 gives an empirical illustration in several representative matrix games by visualizing the PE landscape over the simplex induced by a three-strategy restricted population. The figure shows that the landscape is partitioned into broad regions within which the PE value remains unchanged. This suggests that dispersing meta-strategies over the simplex can increase the chance of hitting regions that induce different BRs, thereby providing a reliable approximation to the full BR space.

### 4.2.2. PHASE II: SELECTION

The selection phase evaluates the candidate populations and chooses the expansion that yields the lowest estimated PE. For each candidate $\Pi_k^+$, we estimate $\mathcal{PE}(\Pi_k^+; \mathcal{G})$ via a best-response–based procedure, which returns (i) an estimated

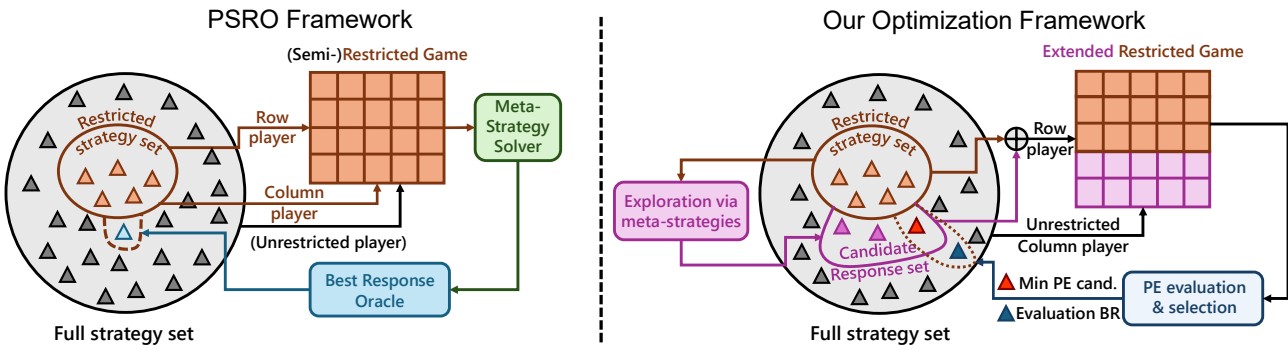

*Figure 2.* Comparison between optimization frameworks. (a) The PSRO framework: an MSS computes a meta-strategy over the current restricted set, and PSRO adds a BR to expand the population. (b) Our framework: each round has two phases. Exploration samples a pool of meta-strategies and trains corresponding responses, forming multiple candidate expansions. Selection estimates each candidate's PE via a best-response–based procedure, picks the candidate with the lowest PE, and adds both the candidate response and the evaluation BR to the population.

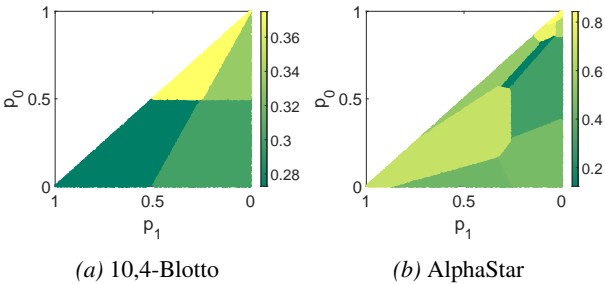

*(a)* 10,4-Blotto       *(b)* AlphaStar

*Figure 3.* Post-expansion PE landscape on the simplex $\{\mathbf{p} \mid p_0 + p_1 + p_2 = 1\}$ induced by a three-strategy restricted population. Additional examples are given in Appendix B.2.

PE value $\widehat{\mathcal{PE}}(\mathbf{\Pi}_k^+; \mathcal{G})$ and (ii) the corresponding response policy trained against the candidate's least-exploitable mixture. We then select

$$k^\star \in \arg\min_{k \in \{1,\dots,K\}} \widehat{\mathcal{PE}}\left(\mathbf{\Pi}_k^+; \mathcal{G}\right), \qquad (9)$$

and expand the population as

$$\mathbf{\Pi}^r \leftarrow \mathbf{\Pi}^r \cup \{\pi_{k^\star}, \beta_{k^\star}\}. \qquad (10)$$

Selecting $k^\star$ uses a *global* post-expansion criterion: we compare candidates by the PE of their *expanded* populations, rather than by restricted-game payoffs. After selecting $k^\star$, we additionally add $\beta_{k^\star}$, the full-game response returned by the PE-evaluation routine against the least-exploitable mixture on $\mathbf{\Pi}_{k^\star}^+$. This is equivalent to an Anytime-PSRO-style expansion applied to the selected population and does not alter the selection criterion.

### 4.2.3. CONVERGENCE ANALYSIS

We analyze the convergence properties of the proposed exploration–selection framework.

**Iteration counting (aligned with PSRO).** We use the standard PSRO counting: one *iteration* means adding *one* new strategy into the restricted population. Our exploration–selection framework adds two strategies per round. Therefore, one framework round corresponds to two PSRO-style iterations under this counting.

**Assumption (Exact oracle).** Whenever the framework queries a BR (either for candidate generation or for PE estimation), the oracle returns an exact BR. Under this assumption, the PE values used for selection are exact.

**Framework conditions.** Fix a base RGB-MSS $\mathcal{M}$. The framework satisfies: (i) *Base inclusion:* the mixture pool always contains the meta-strategy $\sigma^{\mathcal{M}}$ produced by $\mathcal{M}$; and (ii) *Conservative tie-breaking:* if multiple candidates achieve the same minimal PE, we select the candidate induced by $\sigma^{\mathcal{M}}$ whenever it is among the ties.

**Theorem 4.2.** *Assume an exact oracle. Let $\mathcal{M}$ be an RGB-MSS such that PSRO with $\mathcal{M}$ reaches a NE in any two-player zero-sum game in finitely many iterations. Then any instantiation of the exploration–selection framework that satisfies base inclusion and conservative tie-breaking also reaches a NE in finitely many PSRO-style iterations for any two-player zero-sum game.*

The proof of Theorem 4.2 is deferred to Appendix A.4. Theorem 4.2 establishes that the framework preserves the convergence guarantees of RGB-MSS-based PSRO in zero-sum games. A concrete instantiation satisfying the premise is restricted-game NE, for which the corresponding PSRO enjoys finite-iteration convergence in two-player zero-sum games under the exact-oracle assumption.

**Iteration complexity.** Obtaining a non-asymptotic bound on the number of iterations in general games is challenging. Nevertheless, sharper bounds can be derived on adversarial game families previously known to cause worst-case

behavior for RGB-MSS.

**Theorem 4.3.** *Assume the exact oracle. Consider the exploration–selection framework whose exploration pool has size $K$ in each iteration and is formed by including (i) the restricted-game NE and (ii) $K-1$ i.i.d. mixtures sampled from the uniform simplex distribution* Dirichlet($\mathbf{1}$).

*Fix any RGB-MSS $\mathcal{M}$, $N \in \mathbb{N}$, and an odd integer $S$ with $1 \leq S \leq N$. Then there exists an $N \times N$ two-player symmetric zero-sum game $G$ admitting a NE $(\sigma^\star, \sigma^\star)$ with $|\operatorname{supp}(\sigma^\star)| = S$ such that: (i) PSRO instantiated with $\mathcal{M}$ exhibits the worst-case behavior on $G$ (as characterized in Sec. 3.1); yet (ii) running the above exploration–selection framework on $G$ reaches a NE within an expected number of PSRO-style iterations $T^\star$ satisfying*

$$\mathbb{E}[T^\star] \leq \min\left\{2 + \frac{2S}{1-(1-c)^{K-1}},\ N-1\right\}, \quad (11)$$

*where $c \in (0, 1]$ is a game-dependent constant.*

The proof is deferred to Appendix A.5. The theorem is derived from the adversarial family of restricted-game-based PSRO in Sec. 3.1, and shows that enriching exploration and selecting by post-expansion PE can circumvent some worst-case behaviors. In particular, as $K \to \infty$, $(1-c)^{K-1} \to 0$ and the bound approaches $\min\{2 + 2S, N - 1\}$ PSRO-style iterations. Thus, the proposed framework can improve iteration efficiency especially when the equilibrium has a small support; for example, when $S = 1$, the upper bound is at most four iterations.

### 4.3. Global PSRO

We now present **Global PSRO**, a practical instantiation of the exploration–selection framework. A naive implementation would train and evaluate $K$ separate policies per iteration, which is prohibitive in large-scale games. Global PSRO amortizes this cost via *conditional* policies that share parameters across candidates.

#### 4.3.1. EXPLORATION

**Mixture pool and conditional responses.** At each iteration, we first form a pool of meta-strategies $\mathcal{S}_t = \{\sigma_k\}_{k=1}^K \subset \Delta(\mathbf{\Pi}^r)$. We include one mixture produced by a base RGB-MSS $\mathcal{M}$ and sample the remaining mixtures uniformly on the simplex:

$$\mathcal{S}_t = \{\sigma^{\mathcal{M}}\} \cup \{\sigma_k\}_{k=2}^K, \quad \sigma_k \sim \text{Dirichlet}(\mathbf{1}). \quad (12)$$

Given $\mathcal{S}_t$, we train a single conditional explorer $\pi_\theta(a \mid s, \sigma)$ that outputs a response tailored to each $\sigma \in \mathcal{S}_t$:

$$\max_\theta \ J(\theta) = \frac{1}{K} \sum_{k=1}^K U\big(\pi_\theta(\cdot \mid \sigma_k), \sigma_k\big). \quad (13)$$

We then instantiate candidate responses by slicing $\pi_k \triangleq \pi_\theta(\cdot \mid \sigma_k)$, yielding candidate sets $\mathbf{\Pi}_k^+ = \mathbf{\Pi}^r \cup \{\pi_k\}$.

#### 4.3.2. SELECTION

**PE estimation** To select among candidates, we estimate $\mathcal{PE}(\mathbf{\Pi}_k^+; \mathcal{G})$ for each $k$. In two-player zero-sum games,

$$\mathcal{PE}(\mathbf{\Pi}_k^+; \mathcal{G}) = \min_{\sigma \in \Delta(\mathbf{\Pi}_k^+)} \max_{\pi \in \mathbf{\Pi}} U(\pi, \sigma). \quad (14)$$

Exact evaluation of (14) is intractable at scale, so we approximate it using RM–BR (McAleer et al., 2022). For each candidate $k$, RM maintains a mixture $\rho_k \in \Delta(\mathbf{\Pi}_k^+)$ via regret-minimization updates (RM steps), while a best-response learner tracks $\rho_k$ (BR steps). Since $\rho_k$ changes incrementally, we warm-start the BR learner and use a small update budget per BR step.

To avoid training $K$ separate BR learners, we amortize the BR component using a conditional evaluator $\beta_\psi(a \mid s, \sigma)$. For candidate $k$, the BR step is implemented by the slice $\beta_k \triangleq \beta_\psi(\cdot \mid \sigma_k)$, producing the PE estimate $\widehat{\mathcal{PE}}_k = U(\beta_k, \rho_k)$. In our ablations, we isolate the effect of using the estimated PE for selection.

**Regularized evaluation metric.** Because $\rho_k$ and $\beta_k$ are approximate, a candidate may appear to have a small $\widehat{\mathcal{PE}}_k$ even when the newly added strategy $\pi_k$ receives negligible mass under $\rho_k$. Let $\hat{p}_k \triangleq \rho_k(\pi_k)$. We therefore rank candidates by the regularized score

$$\widehat{\mathcal{PE}}_k^{\text{reg}} = (1-\hat{p}_k)\Big(\widehat{\mathcal{PE}}(\mathbf{\Pi}^r; \mathcal{G}) - U(\beta_k, \rho^r)\Big) + \widehat{\mathcal{PE}}_k, \quad (15)$$

where $\rho^r$ is the mixture used to estimate $\widehat{\mathcal{PE}}(\mathbf{\Pi}^r; \mathcal{G})$. Since $U(\beta_k, \rho^r) \leq \widehat{\mathcal{PE}}(\mathbf{\Pi}^r; \mathcal{G})$, the bracketed term is non-negative, so small $\hat{p}_k$ yields a larger upward correction. We also ablate this regularization to quantify its contribution beyond the raw PE estimate.

#### 4.3.3. ALGORITHM SUMMARY

Algorithm 1 summarizes Global PSRO, where $\mathcal{M}(\widehat{\mathbf{U}})$ denotes the meta-strategy produced by base MSS and $\text{EVAL}(\mathbf{\Pi}^r)$ evaluates the empirical payoff table of the restricted population.

## 5. Related Work

Here we briefly highlight two complementary lines of work that connect to our method.

**Diversity-driven response training.** Some PSRO variants encourage exploration by pushing newly added strategies to differ from the existing population, typically by introducing a diversity regularization term into the training objective (Balduzzi et al., 2019; Liu et al., 2022c; Perez-Nieves et al.,

**Algorithm 1** Global PSRO

---

1: **Input:** Initial $\mathbf{\Pi}^r$, explorer $\pi_\theta(\cdot|\sigma)$, evaluator $\beta_\psi(\cdot|\sigma)$, base RGB-MSS $\mathcal{M}$, restricted payoff matrix $\widehat{\mathbf{U}}$
2: **repeat**
3:      // Phase I: Exploration
4:      Form mixture pool $\mathcal{S}_t = \{\sigma_k\}_{k=1}^K$ ($\mathcal{M}(\widehat{\mathbf{U}}) \in \mathcal{S}_t$)
5:      Update Explorer $\pi_\theta$ via DRL against $\{\sigma_k\}_{k=1}^K$
6:      Generate the BR set $\{\pi_\theta(\cdot|\sigma_k)\}_{k=1}^K$
7:      Form the candidate set $\mathbf{\Pi}_k^+ = \mathbf{\Pi}^r \cup \{\pi_\theta(\cdot|\sigma_k)\}$
8:      // Phase II: Selection
9:      Initialize mixed strategy $\rho_k$ for candidate set $\mathbf{\Pi}_k^+$
10:      **for** evaluation steps **do**
11:          Query Evaluator $\beta_k \leftarrow \beta_\psi(\cdot|\sigma_k)$ for all $k$
12:          Generate trajectories $\tau_k$ of $\beta_k$ vs $\rho_k$ for all $k$
13:          Update $\rho_k$ via RM algorithm using $\tau_k$ for all $k$
14:          Update Evaluator $\beta_\psi$ via DRL using $\{\tau_k\}_{k=1}^K$
15:      **end for**
16:      Estimate the PE value $\widehat{\mathrm{PE}}_k^{reg}$ using Eq. 15
17:      Select $k^\star \in \arg\min_k \widehat{\mathrm{PE}}_k^{reg}$, breaking ties in favor of the candidate induced by $\mathcal{M}(\widehat{\mathbf{U}})$
18:      $\mathbf{\Pi}^r \leftarrow \mathbf{\Pi}^r \cup \{\pi_\theta(\cdot|\sigma_{k^*}), \beta_\psi(\cdot|\sigma_{k^*})\}$
19:      $\widehat{\mathbf{U}} \leftarrow \mathrm{EVAL}(\mathbf{\Pi}^r)$
20: **until** Stop condition

---

2021; Liu et al., 2021). A representative example is PSD-PSRO (Yao et al., 2023), which measures diversity via the KL divergence between policies and incorporates it into the training objective. Such methods are complementary to our approach: they can be used to produce more diverse candidate responses in the exploration phase.

**Conditional-population PSRO.** Methods such as NeuPL (Liu et al., 2022b) and Simplex-NeuPL (Liu et al., 2022a) represent the PSRO population with a conditional neural network, enabling parameter sharing across many response policies. They retain PSRO-style population expansion but change how policies are represented and trained. We use conditional models for the same purpose; however, our main contribution is orthogonal: selecting expansions via a post-expansion PE criterion. NeuPL therefore provides a natural comparison for separating gains from parameter sharing and from the selection mechanism.

# 6. Experiments

In this section, we design experiments to answer the following questions: (**RQ1**) Compared with PSRO using different MSS variants, does Global PSRO improve equilibrium approximation? (**RQ2**) How does Global PSRO compare to diversity-driven methods, and can diversity further improve candidate generation within Global PSRO? (**RQ3**) Do the gains of Global PSRO persist when compared to conditional-population methods? (**RQ4**) How much does each component of the framework contribute to the overall performance improvements?

## 6.1. Experimental Setup

**Environments.** We evaluate on widely used two-player zero-sum extensive-form games in the PSRO literature: Kuhn Poker, Liar's Dice, Leduc Poker, and Goofspiel (Appendix C.1). For Goofspiel, we use two variants: a smaller one with 5 cards and a larger one with 13 cards. These benchmarks span a range of scales, from Kuhn Poker to Liar's Dice and Leduc Poker, and finally to large Goofspiel instances.

**Baselines.** We compare *PSRO procedures* under different design choices. (1) **Meta-strategy choices:** PSRO equipped with restricted-game NE (abbreviated as Nash in legends and tables), AlphaRank, PRD, Uniform, and the semi-restricted solver used in Anytime PSRO. (2) **Diversity-driven PSRO:** PSD-PSRO. (3) **Conditional-population PSRO:** NeuPL.

**Interaction budgets.** All methods are compared under the same total number of environment interactions, which is the x-axis in all curves. Each individual policy-training run uses the same number of environment steps. This includes BR training in standard PSRO, candidate-response training in Global PSRO's exploration phase, and RM–BR evaluator training for PE estimation in Global PSRO's selection phase. More details are provided in Appendix C.4.

**Metrics.** We report PE of the learned population as the primary metric. *Note: The PE values shown in all figures are computed solely for evaluation to assess strategy performance, and their interaction costs are not included in the training budget.* For Goofspiel, exact PE evaluation is expensive, so we report a high-budget RM–BR estimate as a proxy. In addition, to align with prior PSRO literature, we also report exploitability, with results in Appendix B.5.

**Implementation details.** All learning-based methods train BRs with PPO (Yu et al., 2022). Extensive-form games are implemented in OpenSpiel (Lanctot et al., 2019). We use a unified policy architecture $\pi_\theta(a \mid s, z)$ with optional conditioning $z$: methods without conditioning fix $z$ to a constant vector, while NeuPL and Global PSRO set $z$ to the mixture vector. Hyperparameters are in Appendix C.5. We report mean and variance over four independent runs.

## 6.2. RQ1: Comparison with PSRO using various MSSs

Figure 4 and Table 1 compare Global PSRO to standard PSRO instantiated with different MSSs under matched environment-step budgets. Global PSRO achieves consistently lower PE on Liar's Dice, Leduc Poker, and all Goofspiel variants, indicating that its learned populations provide a better equilibrium approximation. On Kuhn Poker,

*Table 1.* Estimated PE on Goofspiel (13 cards) across environment-interaction budgets.

| Method | Environment steps ($\times 10^6$) | | | | | | | |
|---|---|---|---|---|---|---|---|---|
| | 19.2 | 38.4 | 57.6 | 76.8 | 96.0 | 115.2 | 134.4 | 153.6 |
| **Global PSRO (ours)** | $0.305 \pm 0.062$ | $\mathbf{0.160 \pm 0.041}$ | $\mathbf{0.150 \pm 0.030}$ | $\mathbf{0.056 \pm 0.037}$ | $\mathbf{0.084 \pm 0.002}$ | $\mathbf{0.079 \pm 0.067}$ | $\mathbf{0.064 \pm 0.017}$ | $\mathbf{0.046 \pm 0.016}$ |
| PSRO w/ Nash | $0.579 \pm 0.086$ | $0.411 \pm 0.062$ | $0.321 \pm 0.040$ | $0.284 \pm 0.043$ | $0.251 \pm 0.023$ | $0.206 \pm 0.016$ | $0.188 \pm 0.014$ | $0.193 \pm 0.044$ |
| PSRO w/ PRD | $0.667 \pm 0.046$ | $0.569 \pm 0.099$ | $0.531 \pm 0.030$ | $0.498 \pm 0.059$ | $0.479 \pm 0.042$ | $0.449 \pm 0.064$ | $0.404 \pm 0.013$ | $0.366 \pm 0.013$ |
| PSRO w/ AlphaRank | $0.459 \pm 0.060$ | $0.292 \pm 0.039$ | $0.221 \pm 0.037$ | $0.188 \pm 0.004$ | $0.170 \pm 0.023$ | $0.162 \pm 0.065$ | $0.190 \pm 0.001$ | $0.178 \pm 0.054$ |
| PSRO w/ Uniform | $0.500 \pm 0.048$ | $0.369 \pm 0.003$ | $0.327 \pm 0.043$ | $0.277 \pm 0.035$ | $0.282 \pm 0.009$ | $0.200 \pm 0.044$ | $0.254 \pm 0.026$ | $0.230 \pm 0.001$ |
| Anytime PSRO | $0.404 \pm 0.048$ | $0.223 \pm 0.083$ | $0.218 \pm 0.083$ | $0.223 \pm 0.096$ | $0.203 \pm 0.060$ | $0.183 \pm 0.058$ | $0.203 \pm 0.056$ | $0.191 \pm 0.061$ |
| NeuPL w/ Nash | $0.337 \pm 0.035$ | $0.298 \pm 0.036$ | $0.234 \pm 0.007$ | $0.225 \pm 0.016$ | $0.182 \pm 0.027$ | $0.162 \pm 0.042$ | $0.169 \pm 0.006$ | $0.142 \pm 0.042$ |
| NeuPL w/ PRD | $0.563 \pm 0.043$ | $0.500 \pm 0.063$ | $0.474 \pm 0.025$ | $0.443 \pm 0.072$ | $0.431 \pm 0.053$ | $0.372 \pm 0.033$ | $0.345 \pm 0.018$ | $0.334 \pm 0.015$ |
| NeuPL w/ AlphaRank | $\mathbf{0.244 \pm 0.011}$ | $0.212 \pm 0.011$ | $0.209 \pm 0.021$ | $0.169 \pm 0.009$ | $0.183 \pm 0.014$ | $0.187 \pm 0.021$ | $0.146 \pm 0.004$ | $0.132 \pm 0.041$ |
| NeuPL w/ Uniform | $0.390 \pm 0.078$ | $0.301 \pm 0.048$ | $0.273 \pm 0.023$ | $0.251 \pm 0.039$ | $0.235 \pm 0.002$ | $0.215 \pm 0.010$ | $0.219 \pm 0.022$ | $0.170 \pm 0.004$ |
| PSD-PSRO w/ AlphaRank | $0.599 \pm 0.010$ | $0.349 \pm 0.021$ | $0.284 \pm 0.009$ | $0.251 \pm 0.003$ | $0.249 \pm 0.029$ | $0.220 \pm 0.006$ | $0.212 \pm 0.003$ | $0.160 \pm 0.006$ |
| PSD-PSRO w/ Nash | $0.670 \pm 0.007$ | $0.494 \pm 0.026$ | $0.396 \pm 0.002$ | $0.335 \pm 0.058$ | $0.306 \pm 0.037$ | $0.282 \pm 0.044$ | $0.258 \pm 0.038$ | $0.211 \pm 0.028$ |
| PSD-PSRO w/ PRD | $0.739 \pm 0.054$ | $0.677 \pm 0.081$ | $0.616 \pm 0.114$ | $0.562 \pm 0.064$ | $0.533 \pm 0.061$ | $0.437 \pm 0.011$ | $0.345 \pm 0.013$ | $0.309 \pm 0.001$ |
| PSD-PSRO w/ Uniform | $0.554 \pm 0.048$ | $0.341 \pm 0.003$ | $0.328 \pm 0.002$ | $0.281 \pm 0.031$ | $0.256 \pm 0.019$ | $0.239 \pm 0.018$ | $0.203 \pm 0.016$ | $0.149 \pm 0.021$ |

restricted-game NE is competitive and can slightly outperform Global PSRO; this is expected in a small game, where restricted games quickly become close to the full game and leveraging additional global information offers limited marginal benefit.

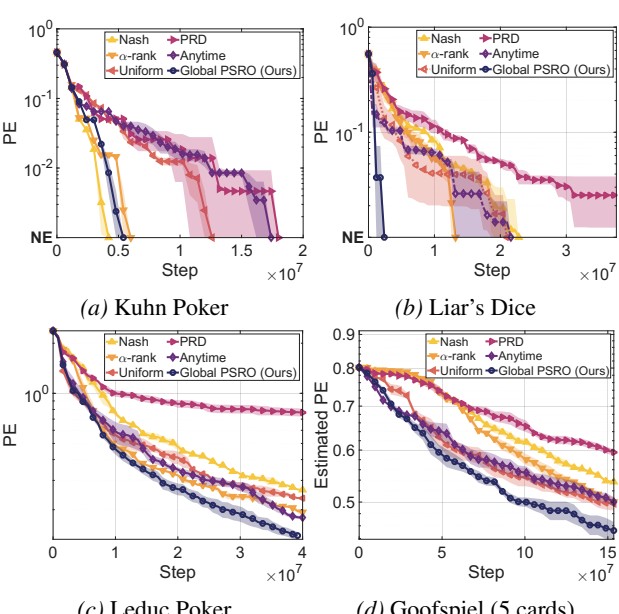

*(a)* Kuhn Poker     *(b)* Liar's Dice

*(c)* Leduc Poker     *(d)* Goofspiel (5 cards)

*Figure 4.* Comparison between Global PSRO and PSRO with different MSSs. Legend entries correspond to the MSS used within PSRO.

### 6.3. RQ2: Comparison with diversity-driven PSRO

Figure 5 compares Global PSRO with the diversity-driven variant PSD-PSRO. Fig. 5 and Table 1 report results on the largest games, including Leduc Poker and all Goofspiel variants; the remaining environments are deferred to Appendix B.3. Under matched budgets, Global PSRO achieves lower PE across games, indicating that favoring diverse additions alone does not reliably produce effective population

expansions.

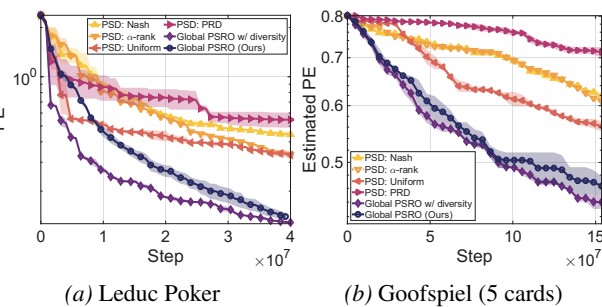

*(a)* Leduc Poker     *(b)* Goofspiel (5 cards)

*Figure 5.* Comparison with diversity-driven PSRO. "Global PSRO w/ diversity" augments the exploration phase with a diversity objective for candidate generation.

We then incorporate diversity into our exploration stage. Specifically, the conditional explorer takes $(\sigma, \lambda)$ as input, where $\lambda \sim \text{Uniform}[0, 0.1]$ weights a diversity regularizer in the BR objective, generating candidates with varying diversity. As shown in Fig. 5, this improves Global PSRO overall, with the largest gains in Leduc Poker and smaller or inconsistent gains in other games.

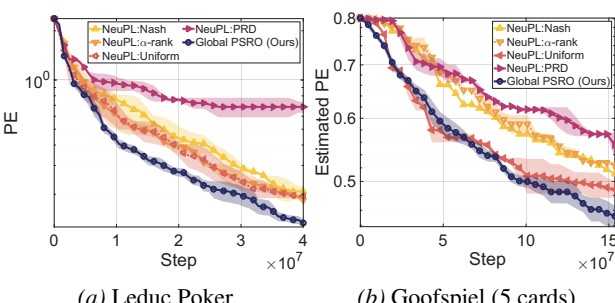

*(a)* Leduc Poker     *(b)* Goofspiel (5 cards)

*Figure 6.* Comparison with NeuPL.

## 6.4. RQ3: Comparison with NeuPL

We compare against NeuPL as a representative PSRO-style baseline that also uses a conditional population model, isolating whether our gains come from the selection mechanism rather than parameter sharing alone. Fig. 6 and Table 1 report results on the largest games, including Leduc Poker and all Goofspiel variants; the remaining environments are deferred to Appendix B.4.

Across these comparisons, Global PSRO achieves lower final PE than all NeuPL variants. This suggests that the improvement is not explained by conditional parameterization alone, but mainly comes from selecting population expansions using the post-expansion PE.

## 6.5. RQ4: Ablation Studies

We ablate key components of the proposed exploration–evaluation pipeline. We consider: (1) *Exploitation only:* the candidate pool contains only the $\sigma^{\mathcal{M}}$ produced by the base RGB-MSS; (2) *Random selection:* generate the same candidate pool as Global PSRO but select a candidate uniformly at random; (3) *Without PE regularization:* rank candidates by the raw estimator $\widehat{\mathcal{PE}}_k = U(\beta_k, \rho_k)$ instead of the regularized score in Eq. (15); (4) *Exact PE evaluation:* replace the estimated PE used for candidate ranking with exact PE values (not counted toward the budget), providing an upper bound on selection quality; and (5) *Neighbor-search exploration:* replace uniform simplex sampling with local perturbations around $\sigma^{\mathcal{M}}$, $\sigma_k = (1 - \gamma)\sigma^{\mathcal{M}} + \gamma\tilde{\sigma}_k$ with $\tilde{\sigma}_k \sim \mathrm{Dirichlet}(\mathbf{1})$.

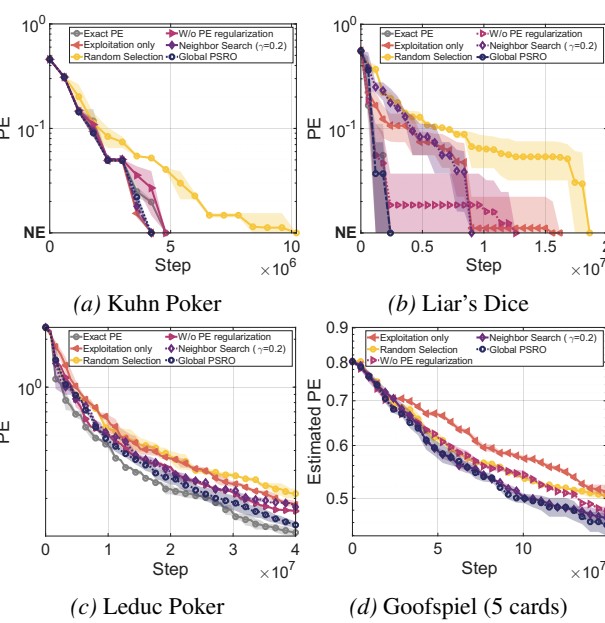

*(a)* Kuhn Poker     *(b)* Liar's Dice

*(c)* Leduc Poker     *(d)* Goofspiel (5 cards)

*Figure 7.* Ablation studies.

Figure 7 reports ablation results across Kuhn Poker, Liar's

Dice, Leduc Poker, and Goofspiel with 5 cards. All ablations degrade performance, confirming that each component contributes to the overall gains. The largest drops come from *exploitation only* and *random selection*, as both effectively disable the mechanism that leverages global information. This supports our main claim that the dominant source of improvement is selecting expansions using global information. Using *exact PE estimation* yields a modest additional gain, suggesting that our practical PE estimator is already informative.

## 7. Conclusions and Discussions

While PSRO-style methods have shown strong empirical performance, strategy population expansions driven solely by restricted-game payoffs may fail to efficiently improve population-level equilibrium approximation. To address this limitation, we propose an exploration–selection framework that uses global information to guide the expansion procedure by evaluating multiple candidates and selecting the strategy that minimizes post-expansion PE, together with its corresponding BR. We instantiate this framework as *Global PSRO*, combining parameter sharing for scalable multi-candidate response generation with PE-based selection. Empirical results demonstrate improved equilibrium approximation over existing PSRO variants under matched interaction budgets, and extensive ablation studies further validate the effectiveness of the proposed framework.

Our discussion in this paper focuses on two-player zero-sum games. Extending the framework to multiplayer general-sum settings is non-trivial because PE no longer has the saddle-point structure in this setting, making it harder to compute. We leave extensions to broader game classes, together with scalable PE approximations or suitable alternative population-level metrics, for future work.

## Impact Statement

This paper presents work whose goal is to advance the field of Machine Learning. There are many potential societal consequences of our work, none which we feel must be specifically highlighted here.

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

# A. Theoretical Analysis

## A.1. Proof of Theorem 3.2

To prove the theorem, we introduce a constructive procedure for filling in the payoff matrix. Fix the RGB-MSS $\mathcal{M}$, the initial strategy $\pi_0$, and the target size $N$. Our goal is to construct a symmetric zero-sum game on pure strategies $\Pi = \{\pi_0, \ldots, \pi_{N-1}\}$ that admits a pure-strategy NE, but for which PSRO with $\mathcal{M}$ cannot reach this equilibrium until the entire strategy set has been incorporated. The construction proceeds by observing the mixtures produced by $\mathcal{M}$ and appending strategies one by one. Each appended strategy is required to satisfy three properties: it is a BR to the current MSS mixture, it was not a BR to any earlier MSS mixture, and it has positive payoff against every previously constructed strategy. If these extensions can be carried out until all $N$ strategies have appeared, then the final strategy strictly beats every other pure strategy and therefore forms a pure-strategy NE in the resulting symmetric zero-sum game. If the construction cannot be continued, then the induced PSRO process has already stopped making effective progress toward any full-game NE, yielding the non-convergence alternative in Theorem 3.2.

The construction uses a simple feasibility argument based on Farkas' lemma.

**Lemma A.1** (Farkas' lemma). *Let $A \in \mathbb{R}^{m \times n}$ and $b \in \mathbb{R}^m$. Exactly one of the following systems is feasible:*

1. *there exists $x \in \mathbb{R}^n$ such that $Ax \leq b$ and $x \geq 0$;*

2. *there exists $y \geq 0$ such that $A^\top y \geq 0$ and $b^\top y < 0$.*

We first introduce the notation used in the construction. At iteration $k$, let $\Pi_k^r$ denote the current restricted population, and let $U_k$ be its restricted payoff matrix. The RGB-MSS outputs a probability vector

$$p_k = \mathcal{M}(U_k),$$

which represents a mixed strategy over $\Pi_k^r$. Since PSRO may reintroduce a pure strategy that is already present, the restricted population may contain duplicate copies of the same underlying strategy. We therefore also use an effective restricted population $\Pi_k^e$, obtained by identifying such duplicates. We write

$$\Pi_k^e = \{\pi_0, \ldots, \pi_{n_k-1}\}$$

for this effective population, $U_k^e$ for the payoff matrix on $\Pi_k^e$, and $p_k^e \in \Delta^{n_k-1}$ for the distribution induced by $p_k$ after aggregating the probabilities assigned to identical pure strategies. Finally, $p_k^E$ denotes a Nash equilibrium strategy of the effective restricted game $U_k^e$.

We initialize the construction by prescribing the first response. In the first PSRO iteration, the restricted population contains only $\pi_0$, so the MSS output is necessarily $p_1 = [1]$. We set $U(\pi_1, \pi_0) = 1$ and, by symmetry and zero-sum structure, $U(\pi_0, \pi_1) = -1$, and take $\pi_1$ as the first BR introduced by PSRO.

Now consider an arbitrary later iteration $k$. For any earlier mixture $p_t^e$ with dimension smaller than $n_k$, we append zeros so that it is evaluated in the current effective game $U_k^e$. For each previous mixture $p_t^e$ with $t < k$, define its current BR value as

$$r_t^{(k)} = \max_{1 \leq i \leq n_k} (U_k^e p_t^e)_i,$$

and set

$$\epsilon_k = \min_{1 \leq t < k} r_t^{(k)}.$$

Thus $\epsilon_k$ is the smallest exploitation level, in the current effective game, among all MSS mixtures generated in previous rounds. Separately, define the current MSS mixture's BR value as

$$e_k = \max_{1 \leq i \leq n_k} (U_k^e p_k^e)_i.$$

The comparison between $e_k$ and $\epsilon_k$ determines the next step of the construction: if $e_k \geq \epsilon_k$, an existing strategy already exploits the current MSS mixture at least as much as the protected threshold, so we can use an existing BR and do not need to introduce a new effective strategy; if $e_k < \epsilon_k$, we will construct a new strategy whose payoff vector preserves all previous BR relations while making it a BR to $p_k^e$.

We prove this formally by induction using two statements.

- $Q_k$: $\epsilon_k > 0$.

- $P_k$: If $e_k < \epsilon_k$, then there exists a new strategy $\pi_{n_k}$ with payoff vector

$$u_k^e = (U(\pi_{n_k}, \pi_0), \ldots, U(\pi_{n_k}, \pi_{n_k-1}))^\top > 0$$

and a margin $\delta' > 0$ satisfying

$$\begin{bmatrix} (p_1^e)^\top \\ \vdots \\ (p_{k-1}^e)^\top \\ -(p_k^e)^\top \\ -(p_k^E)^\top \end{bmatrix} u_k^e \leq \begin{bmatrix} r_1^{(k)} \\ \vdots \\ r_{k-1}^{(k)} \\ -e_k \\ 0 \end{bmatrix} - \delta' \mathbf{1}. \tag{16}$$

The first $k-1$ rows of Eq. 16 ensure that the new strategy is not a BR to any earlier MSS mixture. The row involving $-p_k^e$ ensures that it strictly improves on the current best payoff $e_k$, so it is a BR to the current MSS mixture. The final row ensures that the current restricted-game equilibrium is exploitable by the new strategy, so the current restricted population cannot already contain a full-game NE.

We now show $Q_k \Rightarrow P_k$. Assume $\epsilon_k > 0$ and $e_k < \epsilon_k$. Write

$$u_k^e = x + \delta \mathbf{1}, \qquad x \geq 0,$$

where

$$\delta = \frac{\epsilon_k + e_k}{2}, \qquad \delta' = \frac{\epsilon_k - e_k}{4}.$$

Substituting $u_k^e = x + \delta \mathbf{1}$ into Eq. 16, it is enough to find $x \geq 0$ such that

$$\begin{bmatrix} (p_1^e)^\top \\ \vdots \\ (p_{k-1}^e)^\top \\ -(p_k^e)^\top \\ -(p_k^E)^\top \end{bmatrix} x \leq \begin{bmatrix} r_1^{(k)} - (\delta + \delta') \\ \vdots \\ r_{k-1}^{(k)} - (\delta + \delta') \\ -e_k + (\delta - \delta') \\ \delta - \delta' \end{bmatrix}. \tag{17}$$

For every $t < k$, the definition of $\epsilon_k$ gives $r_t^{(k)} \geq \epsilon_k$. Hence

$$r_t^{(k)} - (\delta + \delta') \geq \epsilon_k - \frac{3\epsilon_k + e_k}{4} > 0.$$

Moreover,

$$-e_k + (\delta - \delta') = \frac{\epsilon_k - e_k}{4} > 0, \qquad \delta - \delta' = \frac{\epsilon_k + 3e_k}{4} \geq 0.$$

Thus the right-hand side of Eq. 17 is nonnegative, and the system is feasible; equivalently, the infeasibility certificate in Lemma A.1 cannot hold. Taking any feasible $x \geq 0$ gives $u_k^e = x + \delta \mathbf{1} > 0$ satisfying Eq. 16, proving $P_k$.

It remains to show $P_k \Rightarrow Q_{k+1}$. There are two cases. If $e_k \geq \epsilon_k$, the current MSS mixture is already exploitable by an existing strategy at least up to the protected threshold. We therefore choose an existing BR and introduce no new effective strategy, so $\epsilon_{k+1} = \epsilon_k > 0$. If $e_k < \epsilon_k$, then by $P_k$ we append a new strategy with $u_k^e > 0$. The updated threshold satisfies

$$\epsilon_{k+1} = \min\{\epsilon_k, (u_k^e)^\top p_k^e\} > 0,$$

because both terms are positive. This completes the induction.

Finally, if the extension case $e_k < \epsilon_k$ occurs enough times to construct all $N$ effective strategies, then the last strategy $\pi_{N-1}$ has strictly positive payoff against every earlier strategy. Since the game is symmetric zero-sum, every earlier strategy has strictly negative payoff against $\pi_{N-1}$, while $U(\pi_{N-1}, \pi_{N-1}) = 0$. Hence $(\pi_{N-1}, \pi_{N-1})$ is a pure-strategy NE of the full game. By construction, this equilibrium strategy is unavailable until all pure strategies have been incorporated, so PSRO requires at least $N - 1$ iterations to reach equilibrium.

If, on the other hand, the effective population cannot be extended for sufficiently many iterations under the mixtures produced by $\mathcal{M}$, then the induced PSRO process never reaches a restricted population containing a full-game NE. To complete the payoff matrix in this case, we continue the construction using the Nash equilibrium of the current restricted game as the surrogate MSS output for the remaining steps. This fills all unspecified payoff entries consistently with the symmetric zero-sum structure, while the original PSRO process driven by $\mathcal{M}$ has already failed to converge. Therefore one of the two alternatives in Theorem 3.2 must hold.

### A.2. Proof of Theorem 3.3

**Roadmap.** The proof proceeds in two parts. Part One proves the special case $S = 1$: we construct a game in which the target RGB-MSS $\mathcal{M}$ still follows the worst-case path, but an instance-specific MSS $\mathcal{M}'$ reaches the pure-strategy NE in three iterations. Part Two extends this construction by replacing the pure equilibrium strategy with an $S$-strategy full-support equilibrium block from a symmetric zero-sum subgame. The strategies in this block have positive payoff against all outside strategies, and their payoffs against outside strategies are chosen to closely mimic the original pure equilibrium strategy. Hence the block remains the equilibrium support of the full game, while the target RGB-MSS still introduces the outside strategies first and only reaches equilibrium after adding the support strategies. At the same time, these $S$ support strategies are exposed as BRs from the three-strategy restricted game, so the shortcut MSS can reach equilibrium within $S + 2$ iterations.

**Code implementation of the following constructive proof.** A concrete version of the construction is provided in the accompanying code.

**Part One: Pure-equilibrium shortcut construction.** Part One is the pure-equilibrium case $S = 1$. We start from the adversarial construction in Theorem 3.2 and focus on the first time when the effective restricted population contains three strategies and the target RGB-MSS produces a mixture that must introduce a new effective response. More precisely, let this iteration be $k$, let the three-strategy core be

$$\Pi_3 = \{\pi_0, \pi_1, \pi_2\},$$

and suppose the MSS mixture satisfies $e_k < \epsilon_k$ in the notation of Theorem 3.2. We then construct the next strategy, denoted by $\pi_3$, with additional constraints. The inherited constraints keep the target RGB-MSS on the same worst-case path, while the new constraints make $\pi_3$ useful for a shortcut MSS defined over the three-strategy core.

Let $u_k^e$ be the payoff vector of $\pi_3$ against the current effective population. As in Theorem 3.2, the first constraints require that $\pi_3$ is not a BR to any earlier MSS mixture, is a BR to the current MSS mixture, and exploits the restricted-game equilibrium. We add one more block of inequalities that forces the payoff of $\pi_3$ against one core strategy to be strictly smaller than its payoff against the other two core strategies. Choose an index $1 < i^* \le 3$ such that some other coordinate of $p_k^e$ is at least as large as $(p_k^e)_{i^*}$, and require

$$U(\pi_3, \pi_{i^*-1}) < U(\pi_3, \pi_{i-1}), \qquad i \ne i^*.$$

Equivalently, define

$$B_{i^*} = \begin{bmatrix} -I_{i^*-1} & \mathbf{1}_{i^*-1} & 0 \\ 0 & \mathbf{1}_{n_k-i^*} & -I_{n_k-i^*} \end{bmatrix}.$$

For any payoff vector $u$, the inequalities $B_{i^*} u \le -\epsilon' \mathbf{1}$ mean that the $i^*$-th coordinate of $u$ is smaller than every other coordinate by margin at least $\epsilon'$. We impose

$$\begin{bmatrix} (p_1^e)^\top \\ \vdots \\ (p_{k-1}^e)^\top \\ -(p_k^e)^\top \\ -(p_k^E)^\top \\ B_{i^*} \end{bmatrix} u_k^e \le \begin{bmatrix} r_1^{(k)} \\ \vdots \\ r_{k-1}^{(k)} \\ -e_k \\ 0 \\ \mathbf{0}_{n_k-1} \end{bmatrix} - \begin{bmatrix} \delta' \mathbf{1}_{n_k+1} \\ \epsilon' \mathbf{1}_{n_k-1} \end{bmatrix},$$

for some margins $\delta', \epsilon' > 0$. This is the same feasibility problem as in Theorem 3.2, with the extra ordering block $B_{i^*}$. We now spell out the feasibility argument.

Write

$$u_k^e = x + \delta b_{i^*}^\alpha, \qquad x \geq 0,$$

where $b_{i^*}^\alpha$ is the all-one vector except that its $i^*$-th coordinate is $\alpha < 1$. Substituting this expression and moving the fixed term $\delta b_{i^*}^\alpha$ to the right-hand side gives the following sufficient system for $x$:

$$\begin{bmatrix} (p_1^e)^\top \\ \vdots \\ (p_{k-1}^e)^\top \\ -(p_k^e)^\top \\ -(p_k^E)^\top \\ B_{i^*} \end{bmatrix} x \leq \begin{bmatrix} r_1^{(k)} - \delta\langle p_1^e, b_{i^*}^\alpha\rangle - \delta' \\ \vdots \\ r_{k-1}^{(k)} - \delta\langle p_{k-1}^e, b_{i^*}^\alpha\rangle - \delta' \\ -e_k + \delta\langle p_k^e, b_{i^*}^\alpha\rangle - \delta' \\ \delta\langle p_k^E, b_{i^*}^\alpha\rangle - \delta' \\ ((1-\alpha)\delta - \epsilon')\mathbf{1}_{n_k-1} \end{bmatrix}.$$

The last block follows from $B_{i^*}b_{i^*}^\alpha = -(1-\alpha)\mathbf{1}_{n_k-1}$.

Choose

$$\delta = \frac{\epsilon_k + e_k}{2}, \qquad \delta' = \frac{\epsilon_k - e_k}{4}, \qquad \alpha = 1 - \frac{\epsilon_k - e_k}{4(\epsilon_k + e_k)}, \qquad \epsilon' = \frac{\epsilon_k - e_k}{16}.$$

We check that every block on the right-hand side is nonnegative. For each previous target-path mixture, $r_t^{(k)} \geq \epsilon_k$ by the definition of $\epsilon_k$, and $\langle p_t^e, b_{i^*}^\alpha\rangle \leq 1$, so

$$r_t^{(k)} - \delta\langle p_t^e, b_{i^*}^\alpha\rangle - \delta' \geq \epsilon_k - \delta - \delta' = \frac{\epsilon_k - e_k}{4} > 0.$$

For the current mixture, the choice of $i^*$ implies $(p_k^e)_{i^*} \leq 1/2$, because some other core coordinate is at least as large. Hence

$$\langle p_k^e, b_{i^*}^\alpha\rangle = 1 - (1-\alpha)(p_k^e)_{i^*} \geq 1 - \frac{1-\alpha}{2},$$

and therefore

$$-e_k + \delta\langle p_k^e, b_{i^*}^\alpha\rangle - \delta' \geq \frac{3}{16}(\epsilon_k - e_k) > 0.$$

For the restricted-game equilibrium vector $p_k^E$, all coordinates of $b_{i^*}^\alpha$ are at least $\alpha$, so

$$\delta\langle p_k^E, b_{i^*}^\alpha\rangle - \delta' \geq \delta\alpha - \delta' = \frac{\epsilon_k + 7e_k}{8} \geq 0.$$

Finally,

$$(1-\alpha)\delta - \epsilon' = \frac{\epsilon_k - e_k}{16} > 0.$$

Thus the induced right-hand side is componentwise nonnegative. By the same Farkas-lemma argument used in Theorem 3.2, there exists $x \geq 0$ satisfying the sufficient system. Consequently $u_k^e = x + \delta b_{i^*}^\alpha > 0$ satisfies all inherited constraints and the additional ordering constraint, and the target RGB-MSS path is not changed.

We now start constructing the shortcut probability vector on the three-strategy core. The goal is to find a distribution over $\Pi_3$ that can later be maintained through the remaining adversarial construction and, in the final step, be made to induce the pure-strategy equilibrium as its BR. The first such candidate is obtained from a zero-sum subgame whose row strategy set is the enlarged effective population $\Pi_{k+1}^e = \Pi_3 \cup \{\pi_3\}$ and whose column strategy set is the core $\Pi_3$. For $q \in \Delta(\Pi_3)$, write

$$\phi_k(q) = \max_{\pi \in \Pi_{k+1}^e} U(\pi, q).$$

Let

$$q_k \in \arg\min_{q \in \Delta(\Pi_3)} \phi_k(q), \qquad v_k = \phi_k(q_k).$$

Thus $q_k$ is the best three-strategy shortcut candidate, and $v_k$ is its exploitability in the current subgame. At this point we need the same two properties as in the original proof:

(1) $\pi_3$ is a best response to $q_k$, i.e.

$$v_k = U(\pi_3, q_k) = \max_{\pi \in \Pi_{k+1}^e} U(\pi, q_k).$$

(2) The shortcut candidate is still less exploitable than the target-path threshold:

$$v_k < \epsilon_{k+1}.$$

For (1), take a saddle point $(\mu_k, q_k)$ of this subgame, where $\mu_k \in \Delta(\Pi_{k+1}^e)$ is the row-player equilibrium strategy. We claim that $\pi_3 \in \text{supp}(\mu_k)$. If not, then $\mu_k$ is supported entirely on $\Pi_3$. Since the subgame restricted to $\Pi_3$ on both sides is symmetric zero-sum, the column player can guarantee row payoff at most zero against any row mixture supported on $\Pi_3$. On the other hand, by construction $U(\pi_3, \pi) > 0$ for every $\pi \in \Pi_3$, so against the column mixture $q_k$ the row player can switch to $\pi_3$ and obtain strictly positive payoff. This contradicts the optimality of a row equilibrium strategy supported only on $\Pi_3$. Hence $\pi_3 \in \text{supp}(\mu_k)$. Every row strategy in the support of a saddle point is a best response to the column equilibrium strategy, so $\pi_3$ is a best response to $q_k$.

For (2), let $\mathcal{P}_k$ denote the three-strategy mixtures from the target path whose exploitability defines $\epsilon_{k+1}$ after $\pi_3$ is added. Equivalently,

$$\epsilon_{k+1} = \min_{q \in \mathcal{P}_k} \phi_k(q).$$

Since $v_k = \min_{q \in \Delta(\Pi_3)} \phi_k(q)$ and $\mathcal{P}_k \subseteq \Delta(\Pi_3)$, we have $v_k \leq \epsilon_{k+1}$, so $v_k > \epsilon_{k+1}$ is impossible. It remains to rule out equality. Suppose $v_k = \epsilon_{k+1}$, and let $q \in \mathcal{P}_k$ attain the minimum above. Then $q$ is also a minimizer of $\phi_k$, hence it is a column equilibrium strategy of the same subgame. By the target-path construction, each mixture in $\mathcal{P}_k$ has a unique BR. Therefore the row equilibrium strategy paired with $q$ must put all mass on this unique BR. However, the argument for (1) applies to every column equilibrium strategy of this subgame, so this unique BR must be $\pi_3$. Thus the equality case can only occur for the current target mixture $p_k^e$, whose unique BR is the newly constructed strategy $\pi_3$.

Now use the column-player optimality condition in the saddle point $(\pi_3, p_k^e)$. If $p_k^e$ is a column equilibrium response to the row pure strategy $\pi_3$, then it must be supported only on minimizers of $U(\pi_3, \cdot)$ over $\Pi_3$. But the ordering constraint enforced by $B_{i^*}$ makes $\pi_{i^*-1}$ the unique minimizer of $U(\pi_3, \cdot)$ on $\Pi_3$, while the choice of $i^*$ guarantees that $p_k^e$ is not supported only on $\pi_{i^*-1}$. This contradiction rules out equality, and therefore $v_k < \epsilon_{k+1}$.

The vector $q_k$ is therefore the starting point of a shortcut path on the three-strategy core. It is not yet the final shortcut vector; rather, the later construction will maintain and possibly update this vector until the last step, where the final shortcut vector is made to directly induce the pure-strategy NE as its BR. The inequality $v_k < \epsilon_{k+1}$ gives the slack needed to preserve this shortcut while the adversarial construction continues.

From this point on, each later stage of the construction maintains two objects. The first is a shortcut probability vector $q_j \in \Delta(\Pi_3)$. The second is its exploitability in the current effective restricted game,

$$v_j := \max_{\pi \in \Pi_j^e} U(\pi, q_j).$$

Starting from the value $v_k$ above, we keep the invariant

$$v_j < \epsilon_{j+1}.$$

This invariant says that the current shortcut vector remains less exploitable than the threshold that protects the target RGB-MSS path, so it can still be used later without disturbing the adversarial construction.

We now describe the continuation of the construction. At a later iteration $j$, let

$$e_j = \max_{\pi \in \Pi_j^e} U(\pi, p_j^e)$$

be the exploitability of the target RGB-MSS mixture in the current effective restricted game, consistent with the notation $e_k$ used above. Also let

$$r_t^{(j)} = \max_{\pi \in \Pi_j^e} U(\pi, p_t^e), \qquad t < j,$$

denote the protected exploitability levels of earlier target-path mixtures. There are three cases.

(1) If $e_j > v_{j-1}$, then an existing strategy is already a sufficient BR to the target RGB-MSS mixture. No new effective strategy is introduced. We keep the shortcut vector unchanged, $q_j = q_{j-1}$, and simply re-evaluate its exploitability in the unchanged effective game, so the invariant is preserved.

(2) If $e_j < v_{j-1}$, or if $e_j = v_{j-1}$ but $p_j^e$ assigns positive probability to some strategy outside the core $\Pi_3$, then we introduce a new outside strategy. In this case the shortcut vector remains unchanged, $q_j = q_{j-1}$, but the new payoff vector $u_j^e$ is constrained not to make this shortcut too exploitable. Concretely, in addition to the inherited target-path constraints, we impose

$$
\begin{bmatrix} (p_1^e)^\top \\ \vdots \\ (p_{j-1}^e)^\top \\ -(p_j^e)^\top \\ -(p_j^E)^\top \\ q_{j-1}^\top \end{bmatrix} u_j^e \leq \begin{bmatrix} r_1^{(j)} \\ \vdots \\ r_{j-1}^{(j)} \\ -\frac{v_{j-1}+\epsilon_j}{2} \\ 0 \\ \frac{v_{j-1}+\epsilon_j}{2} \end{bmatrix} - \bar{\delta}\mathbf{1}.
$$

This is the analogue of the old proof's case (2). To verify feasibility, write

$$
u_j^e = x + \delta_1 \mathbf{1}_{n_j} + \delta_2 c_j, \qquad c_j = \begin{bmatrix} \mathbf{0}_3^\top & \mathbf{1}_{n_j-3}^\top \end{bmatrix}^\top,
$$

and choose

$$
\delta_1 = \frac{\epsilon_j + v_{j-1}}{2} - \frac{(\epsilon_j - v_{j-1})c_j^\top p_j^e}{16}, \qquad \delta_2 = \frac{\epsilon_j - v_{j-1}}{4}, \qquad \bar{\delta} = \frac{(\epsilon_j - v_{j-1})c_j^\top p_j^e}{16}.
$$

Since $v_{j-1} < \epsilon_j$, these choices leave a nonnegative right-hand side for the induced system in $x$, so the same Farkas-lemma argument gives a feasible $u_j^e$. The new strategy obtains payoff below $(v_{j-1}+\epsilon_j)/2$ against $q_{j-1}$, while the target mixture $p_j^e$ is exploited above this midpoint. Hence, after adding the new strategy and re-evaluating the unchanged shortcut vector,

$$
v_j < \frac{v_{j-1} + \epsilon_j}{2} < \epsilon_{j+1}.
$$

(3) The remaining case is that $e_j = v_{j-1}$ and $p_j^e$ is supported entirely on the core $\Pi_3$. In this case we refresh the shortcut vector rather than forcing the old one to remain fixed. The invariant gives the strict slack $v_{j-1} < \epsilon_j$, and hence $e_j < \epsilon_j$. We construct the new payoff vector exactly as in the construction of $\pi_3$, with the same inherited target-path constraints and an ordering block $B_{i^*}$. In addition, we choose the payoff of the new strategy against $p_j^e$ so that, after adding the new strategy, $e_j < r_j^{(j+1)} := \max_{\pi \in \Pi_{j+1}^e} U(\pi, p_j^e) \leq \epsilon_j$. This is feasible by the same Farkas-lemma argument, because $e_j < \epsilon_j$. Thus the new strategy becomes the BR to the target RGB-MSS mixture $p_j^e$, but the new exploitability of this mixture is still below the old threshold. Consequently, $\epsilon_{j+1} = \min\{\epsilon_j, r_j^{(j+1)}\} = r_j^{(j+1)}$.

We then recompute the shortcut vector from the zero-sum subgame whose row strategy set is the updated effective population $\Pi_{j+1}^e$ and whose column strategy set is $\Pi_3$. Define $\phi_j(q) = \max_{\pi \in \Pi_{j+1}^e} U(\pi, q)$ for $q \in \Delta(\Pi_3)$, and choose $q_j \in \arg\min_{q \in \Delta(\Pi_3)} \phi_j(q)$ with $v_j = \phi_j(q_j)$. Since $p_j^e$ itself is a distribution on $\Pi_3$, we immediately have $v_j \leq \phi_j(p_j^e) = r_j^{(j+1)} = \epsilon_{j+1}$. As in the argument immediately after introducing $\pi_3$, the equality case can be ruled out, so

$$
v_j < \epsilon_{j+1}.
$$

We now use the maintained shortcut vector to construct the final pure equilibrium strategy. Let $q_*$ be the shortcut vector available just before the last strategy is added. Let $v_* = \max_{\pi \in \Pi_T^e} U(\pi, q_*)$ be its exploitability in the current effective restricted game, and let $\epsilon_*$ be the corresponding target-path threshold. The invariant maintained above says $v_* < \epsilon_*$. Suppose the last strategy to be constructed is $\pi_{N-1}$, and write $T$ for the corresponding construction step. We construct its payoff vector $u_T^e$ using the same constraints as in Theorem 3.2, but add one extra condition: against the shortcut vector $q_*$, the new final strategy must do better than every strategy already present. Since $v_*$ is exactly the best payoff already available against

$q_*$, this condition is $(u_T^e)^\top q_* > v_*$. Equivalently, we impose the following linear system:

$$
\begin{bmatrix}
(p_1^e)^\top \\
\vdots \\
(p_{T-1}^e)^\top \\
-(p_T^e)^\top \\
-(p_T^E)^\top \\
-q_*^\top
\end{bmatrix}
u_T^e \leq
\begin{bmatrix}
r_1^{(T)} \\
\vdots \\
r_{T-1}^{(T)} \\
-e_T \\
0 \\
-v_*
\end{bmatrix}
- \delta' \mathbf{1}.
$$

The first five blocks are the usual target-path constraints. The last row is the shortcut constraint, because it is equivalent to $(u_T^e)^\top q_* \geq v_* + \delta'$. This extra constraint is feasible by the same Farkas-lemma argument: the strict gap $v_* < \epsilon_*$ gives enough room to include it together with the target-path constraints. Therefore the final strategy $\pi_{N-1}$ is the unique BR to $q_*$. Since the construction also makes $\pi_{N-1}$ strictly beat every earlier pure strategy, $\pi_{N-1}$ is the pure-strategy NE of the full game.

An instance-specific MSS can now reproduce the two mixtures needed to expose the three-strategy core and then output $q_*$. The BR to this third mixture is the final pure-strategy NE, so this MSS reaches equilibrium in three PSRO iterations. This completes the pure-equilibrium shortcut construction.

**Part Two: Mixed-equilibrium extension.** We now extend the pure-strategy equilibrium construction to the case where the target equilibrium has support size $S$. The case $S = 1$ is exactly Part One, so assume that $S > 1$ is odd. We describe the nontrivial regime where the shortcut bound $S + 2$ is active and the inherited three-strategy core is present; when the bound is instead $N - 1$, the statement follows from the trivial enumeration bound.

We inherit the pure-equilibrium game constructed in Part One. More precisely, start from that game with $N - S + 1$ strategies, and let $U_{\text{pure}}(\cdot, \cdot)$ denote its payoff function. Its final strategy, denoted by $\pi_\star$, is a pure-strategy NE. Let $\Pi_{\text{out}} = \Pi_{\text{pure}} \setminus \{\pi_\star\}$ be the set of all other strategies. This inherited game has the following properties. First, $\pi_\star$ strictly beats every outside strategy, i.e., $U_{\text{pure}}(\pi_\star, \rho) > 0$ for every $\rho \in \Pi_{\text{out}}$. Second, the target RGB-MSS is forced to add all strategies in $\Pi_{\text{out}}$ before it can add $\pi_\star$; all inequalities enforcing this path are strict, so they have a positive minimum margin. Third, there is a shortcut MSS that first constructs the three-strategy core $\Pi_3 = \{\pi_0, \pi_1, \pi_2\}$ and then outputs a vector $p \in \Delta(\Pi_3)$ whose BR is exactly the pure equilibrium strategy $\pi_\star$.

We now extend this construction by replacing the single pure equilibrium strategy $\pi_\star$ with a group of strategies $\Pi_{\text{eq}} = \{\pi_1^\star, \ldots, \pi_S^\star\}$. The idea is simple. Internally, the strategies in $\Pi_{\text{eq}}$ form a subgame whose equilibrium has full support on all $S$ strategies and has no smaller equilibrium support. Externally, each strategy in $\Pi_{\text{eq}}$ behaves like the old pure equilibrium strategy $\pi_\star$ against $\Pi_{\text{out}}$, up to arbitrarily small perturbations used later to expose the individual support strategies.

*Remark.* The oddness of $S$ is only due to the skew-symmetric payoff matrix required by symmetric zero-sum games. In a general two-player zero-sum game, the same replacement idea can be implemented for any support size.

Because the strategies in $\Pi_{\text{eq}}$ behave like the old pure equilibrium strategy $\pi_\star$ against $\Pi_{\text{out}}$, this design has two consequences. First, $\Pi_{\text{eq}}$ becomes the equilibrium support of the full game: the support strategies strictly exploit outside strategies, while their internal block supplies the mixed equilibrium. Second, the target RGB-MSS still follows the old outside path and therefore introduces the strategies in $\Pi_{\text{out}}$ first; only after that can it start adding strategies from $\Pi_{\text{eq}}$, and it cannot realize the equilibrium until all of them are present. Thus this path still requires incorporating all $N$ strategies in the constructed game. Formally, the final strategy set is $\Pi_{\text{out}} \cup \Pi_{\text{eq}}$, which has size $(N - S) + S = N$. On $\Pi_{\text{out}} \times \Pi_{\text{out}}$ we copy the old payoff entries from $U_{\text{pure}}$. It remains to define the block on $\Pi_{\text{eq}} \times \Pi_{\text{eq}}$ and the cross payoffs between $\Pi_{\text{eq}}$ and $\Pi_{\text{out}}$.

The internal payoff block is straightforward, so we focus on constructing the payoff between $\Pi_{\text{eq}}$ and $\Pi_{\text{out}}$. A trivial choice would be to make every $\pi_i^\star$ have exactly the same payoff against $\Pi_{\text{out}}$ as the old pure equilibrium strategy $\pi_\star$. But it would also make the support strategies tied from the viewpoint of a shortcut MSS; whether all of them are introduced would then depend on tie-breaking. We instead perturb the payoffs around $U_{\text{pure}}(\pi_\star, \cdot)$ so that three requirements hold: (i) every support strategy still strictly exploits every outside strategy; (ii) these perturbations are small enough that no support strategy is introduced early by the target RGB-MSS; and (iii) there exist $S$ probability vectors on the core $\Pi_3$ whose unique BRs are $\pi_1^\star, \ldots, \pi_S^\star$, respectively.

We first discuss requirement (iii), since this is the part that exposes the individual support strategies from the three-strategy

core. Let

$$u = U_{\mathrm{pure}}(\pi_\star, \Pi_3)$$

be the old payoff vector of the pure equilibrium strategy against the core. The shortcut vector $p \in \Delta(\Pi_3)$ from Part One made $\pi_\star$ the unique BR, so there is a strict margin

$$\gamma = \min_{\rho \in \Pi_{\mathrm{out}}} \left( \langle u, p \rangle - \langle U_{\mathrm{pure}}(\rho, \Pi_3), p \rangle \right) > 0.$$

We say that a probability vector $q_i \in \Delta(\Pi_3)$ exposes $\pi_i^\star$ if $\pi_i^\star$ is the unique BR to $q_i$ in the full game. Our goal is to construct one such vector for each support strategy.

Choose distinct unit tangent directions $h_1, \ldots, h_S$ in the feasible cone of the simplex at $p$; that is, $\mathbf{1}^\top h_i = 0$ and $p + \delta h_i \in \Delta(\Pi_3)$ for all sufficiently small $\delta > 0$. Since the directions are finite and distinct, define

$$\alpha = \min_{i \neq j}(1 - \langle h_i, h_j \rangle) > 0.$$

For small parameters $\delta, \eta > 0$, set

$$q_i = p + \delta h_i, \qquad U(\pi_i^\star, \Pi_3) = u + d_i, \qquad d_i = \eta\big(h_i - \langle p, h_i \rangle \mathbf{1}\big).$$

The perturbation $d_i$ is invisible at the base point $p$, because $\langle p, d_i \rangle = 0$. However, it separates the support strategies at the nearby point $q_i$. For any $j \neq i$,

$$\begin{aligned}
\langle U(\pi_i^\star, \Pi_3), q_i \rangle - \langle U(\pi_j^\star, \Pi_3), q_i \rangle &= \langle d_i - d_j, p + \delta h_i \rangle \\
&= \delta\eta(1 - \langle h_i, h_j \rangle) \\
&\geq \delta\eta\alpha > 0.
\end{aligned}$$

Thus $q_i$ makes $\pi_i^\star$ strictly better than every other support strategy.

It remains, for exposure, to ensure that no outside strategy becomes a BR to $q_i$. This follows from the margin $\gamma$. By taking $\delta$ and $\eta$ sufficiently small, the payoff comparison at $p$ changes only slightly, and hence for every $\rho \in \Pi_{\mathrm{out}}$ and every $i$,

$$\langle U(\pi_i^\star, \Pi_3), q_i \rangle > \langle U(\rho, \Pi_3), q_i \rangle.$$

Combining this inequality with the strict separation among support strategies, each $q_i$ exposes exactly one support strategy, namely $\pi_i^\star$.

The remaining cross-payoff entries can be chosen in the same small-perturbation manner. The core entries have already been set by $U(\pi_i^\star, \Pi_3) = u + d_i$; for $\rho \in \Pi_{\mathrm{out}} \setminus \Pi_3$, take $U(\pi_i^\star, \rho) = U_{\mathrm{pure}}(\pi_\star, \rho) + \xi_{i,\rho}$, and fill all reverse entries by skew-symmetry. Since $U_{\mathrm{pure}}(\pi_\star, \rho) > 0$ for every $\rho \in \Pi_{\mathrm{out}}$, sufficiently small perturbations preserve $U(\pi_i^\star, \rho) > 0$ for all $\pi_i^\star \in \Pi_{\mathrm{eq}}$ and $\rho \in \Pi_{\mathrm{out}}$, so requirement (i) holds. Moreover, the target MSS path in Part One is enforced by strict BR inequalities. Taking $d_i$ and $\xi_{i,\rho}$ smaller than the minimum margin of those inequalities ensures that no support strategy is introduced before the corresponding outside path is completed, so requirement (ii) also holds. Thus the construction has all three desired properties.

Finally define the shortcut MSS $\mathcal{M}'$. It first outputs the same two mixtures as in Part One, thereby exposing the three-strategy core $\Pi_3$. It then outputs

$$q_1, q_2, \ldots, q_S.$$

By the construction above, the corresponding full-game BRs are exactly

$$\pi_1^\star, \pi_2^\star, \ldots, \pi_S^\star.$$

After these $S$ additional iterations, the restricted population contains the entire equilibrium support $\Pi_{\mathrm{eq}}$, and the restricted game already realizes the full-game NE supported on $\Pi_{\mathrm{eq}}$. Hence $\mathcal{M}'$ reaches a NE within $S + 2$ iterations. Together with the trivial bound obtained by enumerating all strategies, this gives $\min\{S + 2, N - 1\}$ and completes the proof of Theorem 3.3.

## A.3. Proof of Proposition 4.1

Let $\Pi^r = \{\pi_1, \ldots, \pi_n\}$, and view any mixture $\sigma \in \Delta(\Pi^r)$ as its probability vector over this ordered set. For any pure strategy $\pi \in \Pi$, define its payoff vector against the restricted population as

$$u(\pi, \Pi^r) = \big(U(\pi, \pi_1), \ldots, U(\pi, \pi_n)\big)^\top.$$

Then the payoff of $\pi$ against a mixture $\sigma$ supported on $\Pi^r$ is

$$U(\pi, \sigma) = \langle u(\pi, \Pi^r), \sigma \rangle.$$

Since $\Pi$ is finite and $\pi^\star$ is the unique BR to $\sigma$, the payoff gap between $\pi^\star$ and its closest competitor is strictly positive:

$$\gamma = \min_{\pi \in \Pi, \ \pi \neq \pi^\star} \langle u(\pi^\star, \Pi^r) - u(\pi, \Pi^r), \sigma \rangle > 0.$$

Also define

$$L = \max_{\pi \in \Pi, \ \pi \neq \pi^\star} \|u(\pi^\star, \Pi^r) - u(\pi, \Pi^r)\|_2.$$

The uniqueness of $\pi^\star$ implies $L > 0$, so we set $\varepsilon = \gamma/(2L)$.

Now take any $\sigma' \in \Delta(\Pi^r)$ with $\|\sigma' - \sigma\|_2 < \varepsilon$. For any $\pi \neq \pi^\star$,

$$
\begin{aligned}
&\langle u(\pi^\star, \Pi^r) - u(\pi, \Pi^r), \sigma' \rangle \\
&= \langle u(\pi^\star, \Pi^r) - u(\pi, \Pi^r), \sigma \rangle + \langle u(\pi^\star, \Pi^r) - u(\pi, \Pi^r), \sigma' - \sigma \rangle \\
&\geq \gamma - \|u(\pi^\star, \Pi^r) - u(\pi, \Pi^r)\|_2 \|\sigma' - \sigma\|_2 \\
&> \gamma - L\varepsilon = \frac{\gamma}{2} > 0.
\end{aligned}
$$

Thus $\pi^\star$ still obtains strictly larger payoff than every $\pi \neq \pi^\star$ against $\sigma'$. Therefore $\pi^\star$ remains the unique BR throughout the neighborhood $\{\sigma' \in \Delta(\Pi^r) : \|\sigma' - \sigma\|_2 < \varepsilon\}$.

## A.4. Proof of Theorem 4.2

We argue by contradiction. Assume the framework does *not* converge on some finite game $\mathcal{G}$, i.e., $\mathcal{PE}(\Pi_t^r; \mathcal{G}) > 0$ for all $t$.

**Step 1: Non-convergence implies eventual stagnation (finite game).** Since the game is finite, $|\Pi| < \infty$. The restricted set $\Pi_t^r$ grows only by adding pure strategies from $\Pi$, so it can change at most $|\Pi| - |\Pi_0^r|$ times. Therefore, there exists an index $t_0$ such that

$$\Pi_t^r = \Pi_{t_0}^r \qquad \text{for all } t \geq t_0.$$

Moreover, by the non-convergence assumption we have $\mathcal{PE}(\Pi_{t_0}^r; \mathcal{G}) > 0$.

**Step 2: On a stagnant suffix, selection reduces to the base MSS.** Fix any $t \geq t_0$. For any candidate update that does not introduce a new strategy, the "expanded" set equals the current set, hence its PE is identical:

$$\mathcal{PE}(\Pi_t^r \cup \{\pi\}; \mathcal{G}) = \mathcal{PE}(\Pi_t^r; \mathcal{G}) \qquad \text{whenever } \pi \in \Pi_t^r.$$

Under the exact oracle, all such no-op candidates attain the same PE value, so selection faces a tie among them. By conservative tie-breaking and base inclusion, the framework must select the candidate induced by $\sigma_t^{\mathcal{M}}$. Thus, on every iteration $t \geq t_0$ in which no new strategy can be added, the framework behaves identically to PSRO instantiated with the base MSS $\mathcal{M}$ on the same restricted set.

**Step 3: Contradiction with the convergence of PSRO($\mathcal{M}$).** Consider running PSRO with MSS $\mathcal{M}$ from the restricted set $\Pi_{t_0}^r$ under the same exact-oracle assumption. By the premise of the theorem, PSRO($\mathcal{M}$) must reach a NE after finitely many PSRO-style iterations. Since $\mathcal{PE}(\Pi_{t_0}^r; \mathcal{G}) > 0$, PSRO($\mathcal{M}$) must introduce a new pure strategy at some future iteration (otherwise the restricted set would remain fixed and PE could not drop to zero). However, Step 2 implies that from $t_0$ onward the framework's tie-breaking forces it to follow the $\mathcal{M}$-induced choice whenever selection does not change the set; combined with the fact that $\Pi_t^r$ is fixed for all $t \geq t_0$, this means even the $\mathcal{M}$-induced update fails to add any new strategy. This contradicts the assumed finite-step convergence of PSRO($\mathcal{M}$) from $\Pi_{t_0}^r$.

Therefore the framework must converge after finitely many PSRO-style iterations.

### A.5. Proof of Theorem 4.3

We need to exhibit a game in the adversarial family for which the exploration–selection framework has the claimed expected iteration bound. We start from the mixed-equilibrium construction in Appendix A.2, and impose three additional generic conditions: (1) the full-game BR space induced by the two-strategy prefix $\{\pi_0, \pi_1\}$ is contained in $\{\pi_1, \pi_2\}$, so the framework deterministically reaches the three-strategy core; (2) letting $r = \min_{\pi_i^\star \in \Pi_{\mathrm{eq}}, \rho \in \Pi_{\mathrm{out}}} U(\pi_i^\star, \rho) > 0$, every entry of the internal payoff block on $\Pi_{\mathrm{eq}} \times \Pi_{\mathrm{eq}}$ has absolute value smaller than $r$; and (3) the subgame induced by $\Pi_{\mathrm{eq}}$, and each subgame induced by a subset of $\Pi_{\mathrm{eq}}$, is nondegenerate, so its Nash equilibrium is unique. Condition (1) can be enforced by setting the payoff of $\pi_2$ against both $\pi_0$ and $\pi_1$ to a common value $c$ close to one, and then keeping all later relevant payoffs strictly below $c$. Condition (2) is obtained by scaling the internal block. Condition (3) is generic for finite games.

We first prove three lemmas that will be used in the iteration bound.

**Lemma A.2.** *For any restricted population $\Pi^r$, if $\Pi^r \subseteq \Pi_{\mathrm{out}}$, then $\mathcal{PE}(\Pi^r; G) \geq r$. If $\Pi^r \cap \Pi_{\mathrm{eq}} \neq \emptyset$, then $\mathcal{PE}(\Pi^r; G) < r$.*

*Proof.* For the first claim, take any mixed strategy $\sigma \in \Delta(\Pi^r)$ with $\Pi^r \subseteq \Pi_{\mathrm{out}}$. For every support strategy $\pi_i^\star \in \Pi_{\mathrm{eq}}$, linearity gives

$$U(\pi_i^\star, \sigma) = \sum_{\rho \in \Pi^r} \sigma(\rho) U(\pi_i^\star, \rho) \geq r.$$

Thus a support strategy provides a unilateral improvement of at least $r$, so $\epsilon(\sigma) \geq r$ for every $\sigma \in \Delta(\Pi^r)$, and $\mathcal{PE}(\Pi^r; G) \geq r$.

For the second claim, suppose $\pi_i^\star \in \Pi^r \cap \Pi_{\mathrm{eq}}$ and consider the pure strategy profile that places all probability on $\pi_i^\star$. Any outside strategy $\rho \in \Pi_{\mathrm{out}}$ obtains negative payoff against $\pi_i^\star$, because $U(\rho, \pi_i^\star) = -U(\pi_i^\star, \rho) < 0$. Any support strategy can gain against $\pi_i^\star$ only through the internal block, whose entries have absolute value strictly smaller than $r$. Therefore the exploitability of this pure profile is smaller than $r$, and $\mathcal{PE}(\Pi^r; G) < r$. $\qquad\square$

**Lemma A.3.** *Suppose $\Pi^r \cap \Pi_{\mathrm{eq}} \neq \emptyset$. Then, for any outside strategy $\rho \in \Pi_{\mathrm{out}}$, adding $\rho$ does not strictly decrease PE:*

$$\mathcal{PE}(\Pi^r \cup \{\rho\}; G) = \mathcal{PE}(\Pi^r; G).$$

*Proof.* Let $E = \Pi^r \cap \Pi_{\mathrm{eq}}$, which is nonempty by assumption. Consider the zero-sum game with row-player strategy set $E$ and column-player strategy set $\Pi_{\mathrm{eq}}$. Let $(x_E, y_E) \in \Delta(E) \times \Delta(\Pi_{\mathrm{eq}})$ be its Nash equilibrium. By the nondegeneracy condition, this equilibrium is unique.

We claim that the same pair $(x_E, y_E)$ also determines $\mathcal{PE}(\Pi^r; G)$ in the zero-sum game with row-player strategy set $\Pi^r$ and column-player strategy set $\Pi$. Indeed, no row strategy in $\Pi^r \setminus E$ can improve against $y_E$, because outside strategies are strictly worse against the equilibrium-support strategies in $\Pi_{\mathrm{eq}}$. Similarly, no column strategy in $\Pi \setminus \Pi_{\mathrm{eq}}$ can improve against $x_E$, because every strategy in $E \subseteq \Pi_{\mathrm{eq}}$ strictly exploits outside strategies. Thus the equilibrium that realizes $\mathcal{PE}(\Pi^r; G)$ uses only $E$ on the restricted side and only $\Pi_{\mathrm{eq}}$ on the full-game side.

Now add an outside strategy $\rho \in \Pi_{\mathrm{out}}$ to the restricted population. The intersection with the equilibrium support is unchanged:

$$(\Pi^r \cup \{\rho\}) \cap \Pi_{\mathrm{eq}} = E.$$

Therefore the same subgame $E$ versus $\Pi_{\mathrm{eq}}$ determines the PE after adding $\rho$, and the PE value is unchanged:

$$\mathcal{PE}(\Pi^r \cup \{\rho\}; G) = \mathcal{PE}(\Pi^r; G).$$

$\qquad\square$

**Lemma A.4.** *Suppose $E = \Pi^r \cap \Pi_{\mathrm{eq}}$ is nonempty and $E \subsetneq \Pi_{\mathrm{eq}}$. Then there exists a missing support strategy $\pi_j^\star \in \Pi_{\mathrm{eq}} \setminus E$ such that*

$$\mathcal{PE}(\Pi^r \cup \{\pi_j^\star\}; G) < \mathcal{PE}(\Pi^r; G).$$

*Proof.* Use the notation from Lemma 2: let $(x_E, y_E)$ be the unique NE of the subgame with restricted side $E$ and full-game side $\Pi_{\mathrm{eq}}$, and let $v_E = \mathcal{PE}(\Pi^r; G)$. Because $E$ is a proper subset of the full equilibrium support, $v_E > 0$.

We use the following PE-improvement criterion. Define the dual guarantee of the full-game side against the current restricted set by

$$g_E(y) = \min_{\pi \in E} U(y, \pi), \qquad y \in \Delta(\Pi_{\text{eq}}).$$

Then $v_E = \max_y g_E(y)$, and by nondegeneracy the maximizer is the unique strategy $y_E$. After adding a missing strategy $\pi_j^\star$, the new PE value is

$$v_j = \max_{y \in \Delta(\Pi_{\text{eq}})} \min\{g_E(y), U(y, \pi_j^\star)\}.$$

This formula gives a simple test for whether adding $\pi_j^\star$ improves the population. Before the addition, the full-game side can guarantee the value $v_E$ by playing $y_E$. After the addition, the restricted side has one more pure strategy available, so $y_E$ continues to certify the old value only if the new strategy is not better against $y_E$ than the old PE value, namely only if $U(y_E, \pi_j^\star) \geq v_E$.

If instead $U(y_E, \pi_j^\star) < v_E$, then the old certificate breaks: at $y_E$ the new minimum in the definition of $v_j$ is already below $v_E$. For any $y$ away from $y_E$, the old guarantee $g_E(y)$ is also below $v_E$, because $y_E$ is the unique maximizer of $g_E$. Hence no full-game-side strategy can still guarantee value $v_E$, and therefore $v_j < v_E$. This is the PE-decrease condition we will use.

It remains to prove that at least one missing strategy satisfies this condition. From $g_E(y_E) = v_E$, we already have

$$U(y_E, \pi) \geq v_E, \qquad \forall \pi \in E.$$

Suppose no missing support strategy satisfies $U(y_E, \pi_j^\star) < v_E$. Then the same lower bound also holds for every $\pi_j^\star \in \Pi_{\text{eq}} \setminus E$, so $U(y_E, \pi) \geq v_E$ for every pure strategy $\pi \in \Pi_{\text{eq}}$. Averaging this inequality with respect to the mixed strategy $y_E$ gives $U(y_E, y_E) \geq v_E > 0$. This is impossible in a symmetric zero-sum game, where every mixed strategy has payoff zero against itself. Therefore some missing strategy $\pi_j^\star \in \Pi_{\text{eq}} \setminus E$ satisfies $U(y_E, \pi_j^\star) < v_E$, and by the criterion above its addition strictly decreases PE. $\qquad\square$

We now apply the lemmas to the exploration–selection process. The argument has three stages: first the construction forces the population to reach a three-strategy core; then PE selection makes the first exposed equilibrium-support strategy preferable to any outside strategy; after that, the same PE-decrease argument repeatedly adds the remaining strategies in $\Pi_{\text{eq}}$ until the full equilibrium support is present.

**Step 1: A three-strategy core is reached deterministically.** We first use the early-core condition stated above. Starting from the initial population $\Pi_0^r = \{\pi_0\}$, under the exact-oracle assumption the first candidate-generation step returns the unique BR $\pi_1$ to $\pi_0$, hence $\pi_1$ is added after one PSRO-style iteration.

Next, we design the payoff structure on the restricted set $\{\pi_0, \pi_1\}$ such that, for the mixture $\rho$ produced by the PE-evaluation subroutine on $\Pi^r = \{\pi_0, \pi_1\}$, the set of BRs in the full game satisfies

$$\mathcal{B}(\rho) \subseteq \{\pi_1, \pi_2\},$$

where $\pi_2 \notin \Pi^r$. Since $\pi_1$ is already contained in the current population, the evaluation best-response added by the framework must introduce $\pi_2$. Therefore, after at most two PSRO-style iterations the restricted population deterministically contains the three-strategy core

$$\Pi^{\text{core}} = \{\pi_0, \pi_1, \pi_2\}.$$

We now use the mixed-equilibrium construction from Appendix A.2. Let

$$\Pi_{\text{eq}} = \{\pi_1^\star, \ldots, \pi_S^\star\}$$

be the equilibrium support and let $\Pi_{\text{out}} = \Pi \setminus \Pi_{\text{eq}}$. After the three-strategy core is present, the construction exposes every support strategy from the core: for each $i \in \{1, \ldots, S\}$, there exists a mixture $q_i \in \Delta(\Pi^{\text{core}})$ such that

$$\mathcal{B}(q_i) = \{\pi_i^\star\}.$$

By local stability of strict best responses in finite games, this remains true in an open neighborhood of $q_i$. Hence, for any population $\Pi^r \supseteq \Pi^{\text{core}}$ that does not yet contain $\pi_i^\star$, the region

$$\mathcal{R}_i(\Pi^r) \triangleq \{\sigma \in \Delta(\Pi^r) : \mathcal{B}(\sigma) = \{\pi_i^\star\}\}$$

has positive Lebesgue measure. Since the game is finite, we can take a uniform lower bound over all such populations and missing support strategies:

$$c \triangleq \inf_{\Pi^r \supseteq \Pi^{\mathrm{core}}, \, \pi_i^\star \notin \Pi^r} \Pr_{\sigma \sim \mathrm{Dirichlet}(\mathbf{1})} [\sigma \in \mathcal{R}_i(\Pi^r)] \in (0, 1].$$

**Step 2: Hitting a missing support strategy.** Fix any state with $\Pi^{\mathrm{core}} \subseteq \Pi_t^r$ and $\Pi_{\mathrm{eq}} \not\subseteq \Pi_t^r$. Choose any missing support strategy $\pi_i^\star \notin \Pi_t^r$. Among the $K - 1$ i.i.d. samples from $\mathrm{Dirichlet}(\mathbf{1})$, the probability that at least one sample lies in $\mathcal{R}_i(\Pi_t^r)$ is at least

$$p \triangleq 1 - (1 - c)^{K-1}.$$

On this event, the exact oracle generates $\pi_i^\star$ as a candidate response.

**Step 3: Collecting the equilibrium support.** After the three-strategy core is present, it remains to add the strategies in the equilibrium support $\Pi_{\mathrm{eq}}$ to the restricted population. The key point is that PE selection prefers progress toward this support whenever such progress is available. If the current population contains no strategy from $\Pi_{\mathrm{eq}}$, Lemma 1 says that adding a support strategy makes PE strictly smaller than adding any strategy from $\Pi_{\mathrm{out}}$. If the current population already contains some, but not all, strategies in $\Pi_{\mathrm{eq}}$, then Lemma 2 says that adding a strategy from $\Pi_{\mathrm{out}}$ cannot strictly decrease PE, while Lemma 3 says that at least one missing strategy from $\Pi_{\mathrm{eq}}$ strictly decreases PE. Thus, whenever the candidate set contains a useful missing support strategy, the PE rule selects a strategy from $\Pi_{\mathrm{eq}}$.

Random exploration determines how long we wait until such a candidate appears. At any state with $\Pi^{\mathrm{core}} \subseteq \Pi_t^r$ and $\Pi_{\mathrm{eq}} \not\subseteq \Pi_t^r$, at least one support strategy is still missing. For each missing support strategy, Step 2 gives probability at least $p$ that the $K - 1$ random samples expose it; hence the probability of exposing some missing support strategy is also at least $p$. Conditional on this event, the PE rule incorporates a previously missing strategy from $\Pi_{\mathrm{eq}}$. Therefore each missing support strategy costs, in expectation, at most $1/p$ framework rounds to discover and add. Since there are at most $S$ support strategies to collect, the expected number of framework rounds after the core is formed is at most $S/p$.

Each framework round adds at most two new strategies: the selected candidate response and the PE-evaluation BR. Hence the expected number of additional PSRO-style iterations after the core is formed is at most $2S/p$. Including the at most two PSRO-style iterations needed to reach the core,

$$\mathbb{E}[T^\star] \leq 2 + \frac{2S}{1 - (1 - c)^{K-1}}.$$

**Step 4: Finite-space bound.** The game has $N$ pure strategies. Thus $T^\star \leq N - 1$ deterministically. Combining this bound with Step 3 yields

$$\mathbb{E}[T^\star] \leq \min\left\{ 2 + \frac{2S}{1 - (1 - c)^{K-1}}, \; N - 1 \right\},$$

which completes the proof.

# B. Additional Results

## B.1. Additional worst-case games

Figure 8 provides additional adversarial game instances for the MSSs considered in the main text.

## B.2. Additional BR distribution on simplex

Figure 9 shows additional simplex visualizations of post-expansion PE regions.

## B.3. Additional results in Diversity-driven PSRO

Figure 10 reports the remaining diversity-driven PSRO comparisons omitted from the main text.

## B.4. Additional results in NeuPL

Figure 11 reports the remaining NeuPL comparisons omitted from the main text.

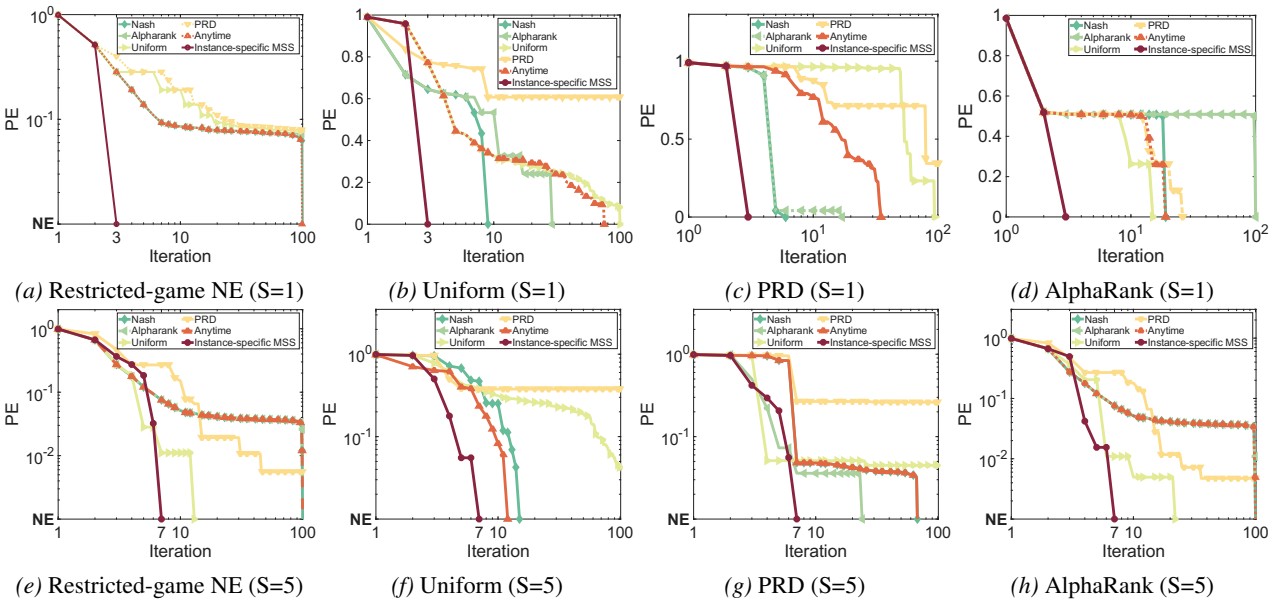

*Figure 8.* PE trajectories of PSRO under various MSSs on adversarially constructed $100 \times 100$ zero-sum games. Each subfigure is tailored to the MSS in its subcaption. "Instance-specific MSS" refers to the MSS $\mathcal{M}'$ from Theorem 3.3 that reaches equilibrium within three iterations.

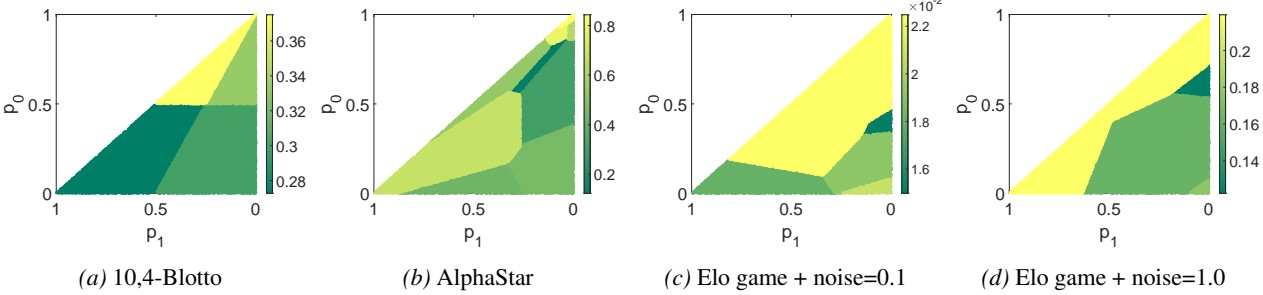

*Figure 9.* Post-expansion PE on simplex $\{\mathbf{p} \mid p_0 + p_1 + p_2 = 1\}$.

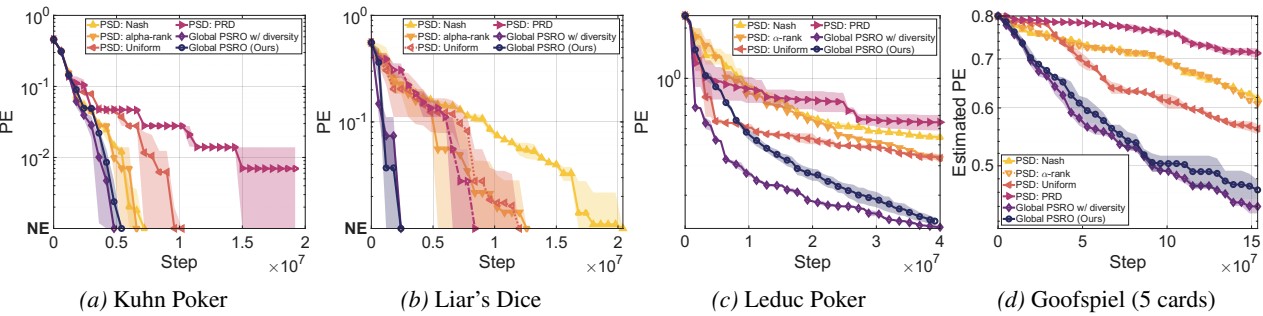

*Figure 10.* Comparison with diversity-driven PSRO. Legends indicate the MSS used by each baseline. "Global PSRO w/ diversity" augments the exploration phase with a diversity objective for candidate generation.

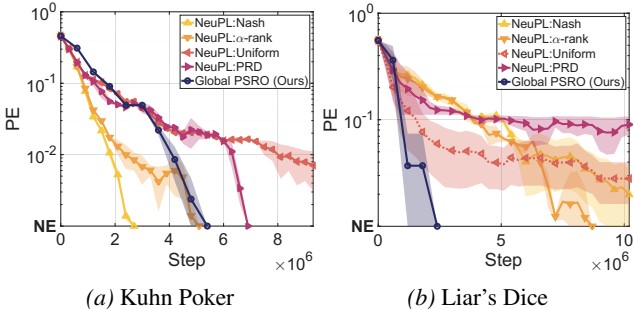

*Figure 11.* Additional NeuPL results.

## B.5. Performance under the Exploitability Metric

We conduct experiments using the exploitability metric employed in vanilla PSRO algorithms. The results are shown in Fig. 12. Global PSRO still maintains the lowest exploitability value after convergence. This is because, in the evaluation phase, Global PSRO uses the regret minimization algorithm to construct a mixed strategy whose exploitability value is close to the PE value of the extended strategy set.

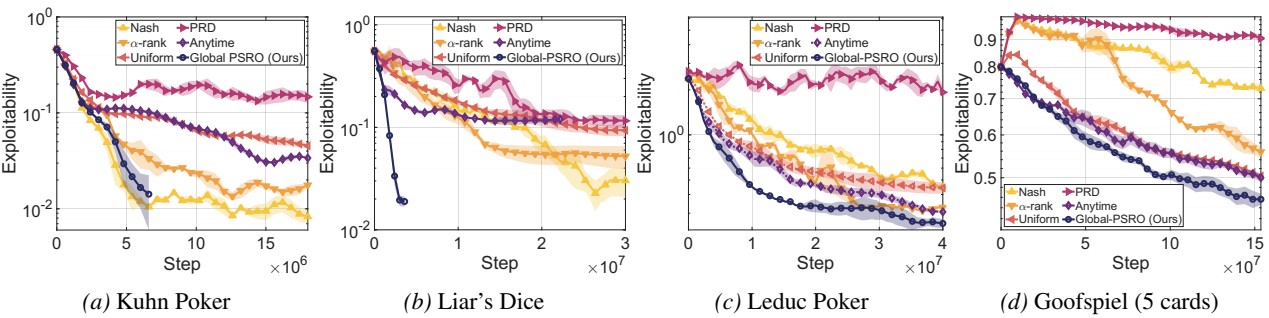

*Figure 12.* Performance under the exploitability metric.

## C. Experimental Details

### C.1. Game Settings

All the extensive-form games used in our PSRO experiments are based on the OpenSpiel library, with the specific parameters as follows:

**Kuhn Poker** Game Name: `kuhn_poker`
    Parameters: `{"players": 2}`

**Liar's Dice** Game Name: `liars_dice`
    Parameters: `{"numdice": 1, "dice_sides": 3}`

**Leduc Poker** Game Name: `leduc_poker`
    Parameters: `{"players": 2}`

**Goofspiel with 5 cards** Game Name: `goofspiel`
    Parameters: `{"num_cards": 5}`

**Goofspiel with 13 cards** Game Name: `goofspiel`
    Parameters: `{"num_cards": 13}`

## C.2. Use of Existing Assets

The primary third-party assets we used are the OpenSpiel library and a standard PPO algorithm library, which are available at https://github.com/google-deepmind/open_spiel and https://github.com/zoeyuchao/mappo, respectively. Detailed information can be found at these links.

## C.3. Hardware Platform

Our hardware platform features an Intel Xeon Gold 6348 CPU @ 2.60GHz, and 8 NVIDIA RTX A6000 GPUs, and 512 GB of memory, running on Ubuntu 18.04.

## C.4. Environment-Interaction Budget

We use environment interaction steps as the common budget unit across all learning-based methods. An environment interaction step corresponds to one rollout transition collected from the game simulator. For standard PSRO baselines, the budget includes rollout steps for BR training and payoff estimation. For Global PSRO, the budget includes rollout steps used for candidate-response training, RM–BR-based PE estimation, and payoff estimation. The RM/Exp3 updates themselves are performed on payoff vectors obtained from rollouts and therefore do not require additional simulator interaction. Candidate generation and PE estimation use shared conditional networks, with rollout budgets allocated at the stage level as reported in the hyperparameter tables.

## C.5. Hyperparameter Settings

Tab. 2 provides the detailed hyperparameters for each algorithm and module in Leduc Poker. Additionally, Tab. 3, 4, 5, and 6 provide the hyperparameters for other games. Only the hyperparameters differing from those in Leduc Poker are included.

*Table 2.* Hyperparameters for training in Leduc poker.

| Algorithm / Module | Name | Value |
|---|---|---|
| Best response oracle | Oracle agent | PPO |
| | Replay buffer size | 800 |
| | Mini-batch size | 2 |
| | Optimizer | Adam |
| | Learning rate | $5 \times 10^{-5}$ |
| | Discount factor ($\gamma$) | 1.0 |
| | Entropy coefficient | 0.04 |
| | Policy network | MLP |
| | Hidden size | 128 |
| | Number of encoder layers | 3 |
| | Number of output layers | 1 |
| | Activation function | ReLU |
| PSRO | Steps for each BR training | $8 \times 10^5$ |
| | Payoff estimation steps per strategy | $4 \times 10^4$ |
| NeuPL | Steps for graph check | $4 \times 10^5$ |
| PSD-PSRO | Steps for each BR training | $8 \times 10^5$ |
| | Diversity weight | 0.1 |
| | Payoff estimation steps per strategy | $4 \times 10^4$ |
| Global PSRO | Steps for each BR training | $8 \times 10^5$ |
| | RM steps of each evaluation phase | 100 |
| | RM algorithm | Exp3 (Auer et al., 1995) |
| | RGB-MSS | Restricted-game NE |
| | Number of workers $K$ | 16 |
| | Payoff estimation steps per strategy | $4 \times 10^4$ |
| Anytime PSRO | Steps for each RM-BR training | $8 \times 10^5$ |
| | RM steps of each RM-BR training | 100 |
| | RM algorithm | Exp3 |

*Table 3.* Hyperparameters in Kuhn Poker.

| Algorithm / Module | Name | Value |
|---|---|---|
| PSRO | Steps for each BR training | $6 \times 10^5$ |
| | Payoff estimation steps per strategy | $3 \times 10^4$ |
| NeuPL | Steps for graph check | $3 \times 10^5$ |
| PSD-PSRO | Steps for each BR training | $6 \times 10^5$ |
| | Diversity weight | 0.05 |
| | Payoff estimation steps per strategy | $3 \times 10^4$ |
| Global PSRO | Steps for each BR training | $6 \times 10^5$ |
| | Payoff estimation steps per strategy | $3 \times 10^4$ |
| Anytime PSRO | Steps for each RM-BR training | $6 \times 10^5$ |

*Table 4.* Hyperparameters in Liar's Dice.

| Algorithm / Module | Name | Value |
|---|---|---|
| PSRO | Steps for each BR training | $6 \times 10^5$ |
| | Payoff estimation steps per strategy | $3 \times 10^4$ |
| NeuPL | Steps for graph check | $3 \times 10^5$ |
| PSD-PSRO | Steps for each BR training | $6 \times 10^5$ |
| | Diversity weight | 0.05 |
| | Payoff estimation steps per strategy | $3 \times 10^4$ |
| Global PSRO | Steps for each BR training | $6 \times 10^5$ |
| | Payoff estimation steps per strategy | $3 \times 10^4$ |
| Anytime PSRO | Steps for each RM-BR training | $6 \times 10^5$ |

*Table 5.* Hyperparameters in Goofspiel with 5 cards.

| Algorithm / Module | Name | Value |
|---|---|---|
| PSRO | Steps for each BR training | $2.4 \times 10^6$ |
| | Payoff estimation steps per strategy | $1.2 \times 10^5$ |
| NeuPL | Steps for graph check | $12 \times 10^5$ |
| PSD-PSRO | Steps for each BR training | $24 \times 10^5$ |
| | Diversity weight | 0.05 |
| | Payoff estimation steps per strategy | $1.2 \times 10^5$ |
| Global PSRO | Steps for each BR training | $2.4 \times 10^6$ |
| | RM steps of each evaluation phase | 300 |
| | Payoff estimation steps per strategy | $1.2 \times 10^5$ |
| Anytime PSRO | Steps for each RM-BR training | $2.4 \times 10^6$ |
| | RM steps of each RM-BR training | 300 |

*Table 6.* Hyperparameters in Goofspiel with 13 cards.

| Algorithm / Module | Name | Value |
|---|---|---|
| PSRO | Steps for each BR training | $2.4 \times 10^6$ |
| | Payoff estimation steps per strategy | $1.2 \times 10^5$ |
| NeuPL | Steps for graph check | $1.2 \times 10^6$ |
| | Payoff estimation steps per strategy | $1.2 \times 10^5$ |
| PSD-PSRO | Steps for each BR training | $2.4 \times 10^6$ |
| | Diversity weight | 0.03 |
| | Payoff estimation steps per strategy | $1.2 \times 10^5$ |
| Global PSRO | Steps for BR training in exploration | $2.4 \times 10^6$ |
| | Steps for RM-BR training in selection | $2.4 \times 10^6$ |
| | Payoff estimation steps per strategy | $1.2 \times 10^5$ |
| Anytime PSRO | Steps for each RM-BR training | $2.4 \times 10^6$ |

