# OpenReview forum: "Global Policy-Space Response Oracles for Two-Player Zero-Sum Games"
_ICML.cc/2026/Conference — ICML 2026 regular_

### Official Review · Reviewer_H62N · 2026-03-10

**Soundness:** 4
**Presentation:** 4
**Significance:** 3
**Originality:** 3
**Overall Recommendation:** 5
**Confidence:** 4

**Summary:**

This paper presents a new method to solve large zero-sum games. The method presents an advance in the PSRO framework, where a small population of strategies are maintained and updated such that the game on the smaller population represents the true game accurately. The paper presents two major contributions in my opinion: (i) A new characterization of PSRO methods based on how much of the true game payoff information does a method use while updating the (smaller) population strategy set, and (ii) presenting a new population expansion method, posing population expansion as a two stage problem aiming to optimizing population exploitability of the updated population strategy set. A third contribution is the global PSRO algorithm, which is a practical deep RL based algorithm which does the two stage population expansion in a manner which is tractable for large zero-sum games. The paper demonstrates favorable performance of global PSRO on numerous large scale zero sum games empirically, which the characterization which is developed to analyze PSRO frameworks theoretically show the benefits of the two stage expansion framework and the pitfalls of the methods which ignore true game payoff information during population expansion.

**Compliance With Llm Reviewing Policy:**

Affirmed.

**Final Justification:**

After reading the author responses to my and other reviewers' questions, I am happy to maintain my current score. Particularly, after reading the discussion between authors and other reviewers:
* I agree that CFR based comparison may not be appropriate as it is not directly comparable here (reviewer sNb5).
* I am satisfied with a paper highlighting specific failure modes and not providing larger theoretical results/guarantees etc (reviewer DLt3).

**Key Questions For Authors:**

1. Can the authors provide some insight into how the convergence to NE is affected by tie breaking scheme in Algorithm 1 (say, with respect to, arbitrary tie breaking)?
2. Do the authors envision that over the course of population expansion, as one edges towards a NE, the size of the mixture pool for introducing new population members can be reduced, eg., through some annealing?

**Limitations:**

yes

**Strengths And Weaknesses:**

Strengths:
1) The biggest strength in my opinion is the new characterization of PSRO on the basis of true game payoff information during population expansion which this paper introduces. This characterization not only allows the authors to design the favorably performing global PSRO, but will also help guide future strong works in this area of research in my opinion.
2) I find the paper to be quite well written, and the claims to be well supported by theory and empirical evidence.

Weaknesses:
1) I think the choice the size of the pool of mixture pool ($K$) can be difficult to tune for an arbitrary zero sum game. Although to be fair to the paper, some tuning would be required for any possible exploration scheme.

---

> ### Author Rebuttal · Authors · 2026-03-31
>
> Thank you for the positive assessment and for highlighting the value of our global-information perspective on PSRO. We are glad that the reviewer found the paper well written and the claims well supported by both theory and experiments.
>
> **Tie-breaking scheme.** Conservative tie-breaking is mainly used to obtain the deterministic convergence result in Theorem 4.2. In tie cases, it keeps the update aligned with the base MSS, which allows the framework to inherit the convergence guarantee of the underlying PSRO. If ties were broken arbitrarily (e.g., randomly), we expect that the guarantee may weaken to a probabilistic one, but establishing such a result would require additional analysis. In practice, exact ties are rare, so this issue mainly matters for the theoretical analysis.
>
> **The mixture-pool size K.** We agree that an adaptive or annealed schedule for K is a promising direction. Intuitively, a larger K encourages broader exploration in earlier stages, while a smaller K may suffice as the population approaches equilibrium. We leave a more systematic investigation of such adaptive strategies to future work.

---

> > ### Author Rebuttal · Reviewer_H62N · 2026-04-03
> >
> > Thank you for your response. I retain my score.

---

> > > ### Author Response · Authors · 2026-04-08
> > >
> > > Thank you for your acknowledgement and for maintaining your positive score.

---

### Official Review · Reviewer_A6yt · 2026-03-11

**Soundness:** 2
**Presentation:** 2
**Significance:** 2
**Originality:** 3
**Overall Recommendation:** 3
**Confidence:** 4

**Summary:**

This paper studies PSRO for two-player zero-sum games, a framework designed to approximate Nash equilibria in large-scale multi-agent games. The authors argue that most standard PSRO algorithms fall short in expanding the restricted set of strategies in a meaningful manner so as to achieve global guarantees. According to the paper, this is due to a fundamental deficiency in the way they evaluate the meta-strategies, i.e., those to be included after the current iteration. Instead, they propose an alternative method, named Global PSRO, which aims to address this issue.

**Compliance With Llm Reviewing Policy:**

Affirmed.

**Final Justification:**

After reading the authors’ responses to my questions and those of the other reviewers, I have decided to maintain my current score. My initial concerns regarding the exploration step of the algorithm were not addressed directly by the authors. In particular, my concern is that what they describe as an exploration step appears instead to be an arbitrary way of populating the current strategy set, without any theoretical guarantee that doing so will improve the algorithm.

**Key Questions For Authors:**

My questions for the authors are essentially the main points raised in the weaknesses section above.

**Limitations:**

The paper does not highlight any limitation or societal impact, but I do not see any specific negative societal impact here.

**Strengths And Weaknesses:**

**Strengths**

Even though I am not very familiar with the whole literature on the PSRO paradigm, I found the overall motivation of the paper sensible. In particular, the authors identify a potential limitation of standard PSRO frameworks, namely that the evaluation of the next meta-strategy depends only on the current population, i.e., the restricted strategy set. According to the authors, this may prevent such methods from achieving global guarantees. In this respect, the main idea of the paper is natural: instead of evaluating a candidate strategy only with respect to the current population, one should evaluate it based on the effect it has on the post-expansion population, measured through the PE (population exploitability) metric. Although I still have several doubts about the technical development and the overall justification, I think the high-level motivation is reasonable and potentially interesting.

**Weaknesses**

I will try to present the points in the way I encountered them. First, regarding the PE objective introduced in this work, the authors state that a “commonly-used” population-level quantity is PE (population exploitability). However, in [1], for example, population exploitability is defined differently (somewhat similar in Equation (6)).

Starting with Section 3.1, there is no clean comparison with previous works to evaluate the contribution of this work. Specifically, the authors talk about “restricted-game Nash, projected replicator dynamics (PRD), AlphaRank, and Uniform” without citing existing works; also, for the term “Uniform,” it is not understood what exactly it stands for. Also, what is the connection with “AlphaRank”? As far as I know, it seems unrelated to the scope of the paper. The same applies to the term “restricted-game Nash”; this is not a “representative example”, as the authors claim for a PSRO instantiation. This is simply a term. Again, regarding “projected replicator dynamics (PRD),” which is mainly a continuous-time algorithm, how does it constitute a representative example of a PSRO instantiation? Without trying to sound strict, the whole subsection gives me the idea that the authors really wanted to enumerate some terms. Apart from that, my main concern is why one should really consider the Restricted-Game-Based PSRo framework, as it is quite obvious that if the meta-strategy is evaluated only based on the restricted payoff game, then how is it possible to expand the population in any meaningful manner? It is not clear from the Section, whether we should either consider them. However, if there are previous work that do that, then there should be a reason that the authors should mention.

Now, the primary concern I have is regarding the way the next strategy is selected, as introduced in Section 4, namely the main idea of the paper. First, even though, as argued above, it makes sense for the best responses to be allowed to come from the full game, then the definition in (6) may consider every policy in the full game. If this is the case, then by enumerating all strategies, the procedure will eventually find the Nash equilibrium. Of course, the authors themselves note this limitation, and for this reason they propose an alternative (approximate) procedure to ameliorate this problem. However, I do not see how this is true at all. There is no condition on how candidate strategies are even selected so that they can reflect *global* guarantees, which contradicts their fundamental motivation for proposing such a framework. Specifically, they claim that the “exploration phase aims to diversify the meta-strategies in $S_t$​ as much as possible,” but it seems that the candidate set is simply a random set without any intrinsic property related to the global guarantees the authors want to achieve. For instance, even considering candidates out of $\mathcal{B}(\Delta(\Pi^r))$ would be more sensible, but that would still have other issues.

Finally, I did not really understand the significance of the finite-iteration convergence claim. Since the game has finitely many actions, even if exponentially many, finite-time convergence under exact oracle assumptions feels close to trivial.


Comments

- In the Introduction of the paper, the authors talk about imperfect-information games and cite papers about imperfect-information games, but the paper is about normal-form games. Is this not somewhat inconsistent with the scope of the paper?

- Definition 3.1: There is an inconsistency. In the definition of RGB-MSS they use $n$ for the number of strategies, whereas initially $n$ denoted the number of players.

- Line 220 (second column): How is $\sigma^{\mathcal{M}}$ defined?

- In Section 3.2, where the SRGM is introduced, the definition of exploitability is wrong. Intuitively, it should capture the worst-case utility, but $\Pi^r$ and $\Pi$ are both defined over both players’ actions, so the utility is not properly defined (see definition in [2])

[1] Policy Space Diversity for Non-Transitive Games

[2] Anytime PSRO for two-player zero-sum games

---

> ### Author Rebuttal · Authors · 2026-03-31
>
> Thank you for the careful review.
>
> **Clarifying the standard PSRO update.**
> As noted in the review, ``if the meta-strategy is evaluated only based on the restricted payoff game, then how is it possible to expand the population in any meaningful manner?'' Standard PSRO does not add the meta-strategy itself to the population. Instead, for each player, it adds a best response to the opponent's meta-strategy into that player's restricted population. Because this best response usually lies outside the current restricted population, adding it yields a nontrivial expansion.
>
> **Clarifying the role of PRD / AlphaRank / Uniform / restricted-game Nash.**
> Our comparison here is at the level of the MSS, i.e., the part of PSRO that compute the meta-strategies of the current restricted game. In this sense, ``restricted-game Nash''[1,2] means using a Nash equilibrium of the current restricted game as the meta-strategy; PRD [2] and AlphaRank [3] mean using, respectively, the mixed strategy produced by PRD and the distribution induced by AlphaRank on the restricted game. and Uniform [4] means using the uniform distribution over the current restricted population. Thus, Sec.~3.1 compares PSRO instantiated with different MSS choices. We will make this abstraction level explicit, define Uniform concretely, and add the missing citations in the revision.
>
> **Why we consider restricted-game-based PSRO.**
> As noted in the review, ``my main concern is why one should really consider the Restricted-Game-Based PSRO framework....'' Our goal in Section 3 is to analyze PSRO from the perspective of information utilization, in order to identify common weaknesses in existing methods. From this viewpoint, it is natural to focus on \emph{restricted-game-based PSRO}, where the meta-strategy is computed solely from the payoff matrix of the current restricted game. Seen in this way, these methods share a common issue: relying only on restricted-game information may lead to inefficient population expansion. This is the focus of Sec. 3.1.
>
> **Clarification on Eq.~(6) and candidate generation.**
> As noted in the review, the concerns here are both that "the definition in (6) may consider every policy in the full game'' and that "the candidate set is simply a random set without any intrinsic property related to the global guarantees....'' Eq. (6) is the ideal objective for a single iteration in our framework, and the full population is obtained by applying such steps iteratively. Such a step is not taken over arbitrary policies in the full game; rather, it is taken over the best responses induced by meta-strategies over the current restricted population, namely $B(\Delta(\Pi^r))$. The candidates are sampled from $B(\Delta(\Pi^r))$, while their global effectiveness is evaluated in selection phase, where candidate expansions are ranked by the PE of their post-expansion populations.
>
> **Scope of the theoretical guarantee.**
> As noted in the review, "I did not really understand the significance of the finite-iteration convergence claim....'' Since Global PSRO uses RL as an approximation to the exact oracle, it is important to first understand the basic property of the framework in that idealized case. More complex settings, in particular the non-exact-oracle case, require a more refined analysis and are beyond the scope of the current paper; we view this as an important direction for future work.
>
> **On PE and prior definitions.**
> Regarding the comment that our PE differs from [5] ([1] in the review), our definition is the general form. In the two-player zero-sum setting, it is consistent with the PE notion used in prior work. We will include a detailed formula derivation in the revised appendix to make this correspondence explicit.
>
> **Comment 1. Scope.**
> The imperfect-information games mentioned in the introduction are the application domains we study, while the theoretical analysis is carried out using their normal-form abstraction. We will make this relationship between the application domains and the analysis level more explicit earlier in the paper.
>
> **Comment 2. Definition 3.1.**
> We will fix this notation inconsistency in the revision.
>
> **Comment 3. Definition of $\sigma^{\mathcal M}$.**
> $\sigma^{\mathcal M}$.denotes the meta-strategy produced by the MSS $\mathcal{M}$, where $\mathcal M$ is the base RGB-MSS used in the Global PSRO.
>
> **Comment 4. Exploitability in Sec. 3.2.**
> In Secs.~3.1--3.2, we switch to a symmetric two-player zero-sum shorthand and suppress player indices. We will rewrite this part more explicitly so that the correspondence is clear.
>
> [1] Planning in the presence of cost functions controlled by an adversary.
>
> [2] A unified game-theoretic approach to multiagent reinforcement learning.
>
> [3] A generalized training approach for multiagent learning.
>
> [4] Deep reinforcement learning from self-play in imperfect-information games.
>
> [5] Policy space diversity for non-transitive games.

---

> > ### Author Rebuttal · Reviewer_A6yt · 2026-04-02
> >
> > Here is a polished version that keeps your point but improves clarity and tone:
> >
> > I thank the authors for their response. However, my main original concern was not addressed, which is why I selected (b) in the Acknowledgement section.
> >
> > The authors state that “Eq. (6) is the ideal objective for a single iteration in our framework, and the full population is obtained by applying such steps iteratively.” This is, of course, a best-case scenario, and as I already noted in my review, it is clearly not achievable in large games.
> >
> > The authors further state that “rather, it is taken over the best responses induced by meta-strategies over the current restricted population.” The issue, as I explained in my review, is that the pool of meta-strategies is entirely arbitrary. As a result, the best responses to those selected for inclusion in the population do not necessarily have any *value*. As I noted in my original review:
> > > but it seems that the candidate set is simply a random set without any intrinsic property related to the global guarantees the authors want to achieve.

---

> > > ### Author Response · Authors · 2026-04-07
> > >
> > > Thank you for the follow-up. As noted in your comments, Eq. (6), our ideal one-step objective, is not directly tractable in large games. In other words, we cannot directly know which candidate is globally best. Our practical approximation therefore uses two steps. In the exploration step, we generate a broad and diverse candidate set of best responses to meta-strategies over the current restricted strategy set, so as to reduce the chance of missing candidates that may be globally strong. The PE-based selection step then performs the global evaluation over these candidates and chooses the most promising expansion. Thus, exploration is intended to preserve coverage of potentially strong candidates, rather than requiring each generated candidate to already be globally good.
> > >
> > > To achieve this exploration goal, we use randomly sampled meta-strategies to generate candidate responses. Here, “random” does not mean that the meta-strategies are entirely arbitrary: each meta-strategy is a probability mixture over the restricted strategy set, and “random” simply refers to sampling its probability vector on the simplex. This exploration scheme relies on a key intrinsic property, formalized in Theorem 4.1: the same best response is often shared by a surrounding region of meta-strategies. Under this structure, finite random samples can provide meaningful, though probabilistic rather than exhaustive, coverage of multiple best-response regions while remaining tractable compared with enumerating the simplex of meta-strategies. We also compared against non-random exploration variants in our ablation study, including “Exploitation only” and “Neighbor-search exploration”. These variants restrict exploration to more limited parts of the simplex, thereby reducing coverage over best-response regions, and both perform worse than our random sampling scheme.

---

### Official Review · Reviewer_DLt3 · 2026-03-11

**Soundness:** 2
**Presentation:** 3
**Significance:** 2
**Originality:** 3
**Overall Recommendation:** 4
**Confidence:** 3

**Summary:**

The paper presents an extension of policy-space response oracles (PSRO) for two-player zero-sum games. In particular, they propose guiding the expansion of the subgame by taking the best-response strategy that minimizes the population exploitability (PE) directly. This is different from prior PSRO variants as it incorporates more information from the full game in the update step. The authors provide some theoretical statements about the relation of their method to previous, more restricted variants and perform empirical experiments on several standard tasks. They compare against previous methods and provide ablation studies.

**Compliance With Llm Reviewing Policy:**

Affirmed.

**Final Justification:**

I repeat my answer of my acknowledgment here:

I increased my score again due to the latest author rebuttal. While I still see several points to be improved, I think the paper is within the acceptance range.

**Key Questions For Authors:**

1. Can you provide a more thorough description of how you count "steps" for your experiments? An example of one RBG-MSS case and the global PSRO one would be very helpful.
2. Are you going to publish your code?

**Limitations:**

yes

**Strengths And Weaknesses:**

The paper presents an intuitive extension of guiding the subgame’s expansion more effectively. In general, I like methods that formulate the underlying problem to be solved first – however difficult it may be – and start from there. As in this case, the optimal greedy expansion is the solution of Equation 6. The authors then construct a feasible method to approximate the solution to Equation 6. They give some theoretical insights into why their extension should be a strict improvement (in some sense, see below for more). Furthermore, they perform some empirical experiments that show the feasibility and improvement of their method. That is, I like the overall approach and the paper’s structure. However, I have several concerns and potential misunderstandings that need to be cleared up for me before publication.

My main concern regards the question how confidently we can say that this method is an improvement to previous ones. I see three major issues why this work might have very limited impact.

## Interpretation of empirical results

The first relates to the interpretation of the empirical results. The method uses previous MSS strategies to select one of the candidates within the optimization of Equation 6 and always adds its best response to the larger game for the next step. So, if we set K=1, then this is the only thing that remains and we are back to the original MSS algorithm. Therefore, in terms of iterations, the new algorithm has to perform not worse than the base MSS algorithm. That being said, one needs to pay a lot of attention trying to make comparisons. Here are my concerns and potential misunderstandings:
-	I am confused what the “step” in the plot really entails and what is a fair comparison here. That is important, as the pure iteration number of the algorithm may not say much (see above). The short paragraph “Computation Budgets” in Section 6.1 does not discuss this in a sufficient manner in my opinion. I would assume that the total number of iterations (that is game expansions) is relatively small, maybe in the order of a couple of hundred. The total number of “steps” lies high in the order of 2-15*1e7. When the authors speak about aligning cost by using the same step budget for (i) BR, (ii) exploration phase, and (iii) PE estimation, then I do not know what that means. The number of steps for RM steps for each evaluation phase is only 100. Does that mean that this is only 100/(8*1e5) of the costs of the BR training or are both things counted as a single step? The number of workers K is set to 16. The learned conditional network leverages generalization capacities from different strategies. Does the number of samples seen by the network scale linearly with K, then I would expect that for K >>16 the step curve looks much worse for global PSRO because one has diminishing returns of more strategies to select from in terms of the costs invested. However, if the number of samples does not scale linearly with K, then this might skew the results even more as the cost of testing K different candidates is not considered. (If we simply choose one at random instead of computing a BR, such as orange line in Figure 5, should the entire line then not be shifted to the left?) Overall, I want to emphasize that I cannot confidently say that I understand whether the proposed method is actually beneficial compared to previous methods when considering computational costs (also the size of the subgame is important here – I expand two actions per iteration instead of only one).
-	In Table 1, you set the RGB-MSS of global PSRO to Restricted-game Nash in Leduc Poker. Why is it this one and not another? It is not clear to me which one was used for the other games.

## Limitation of theoretical results

The second issue is regarding the theoretical results. The statements of Theorems 3.2 and 3.3 are very limited. Theorem 3.2 states the worst-case scenario for RBG-MSS algorithms that they may need to expand the entire game (which is well known for most of them). However, the worst-case scenario is restricted to the construction of games that admit a pure strategy NE. However, finding a pure strategy NE is known to be easy in general. The intuition that the authors want to convey here, that RGB-MSS algorithms are inherently limited and that a general MSS algorithm (Theorem 3.3) may resolve this, is also limited to pure NE for very specific games. That is not a general observation or convergence result. That is, Theorem 3.2 is a general observation (valid but limited) and Theorem 3.3 is an improvement on only these very specific games. For example, the more relevant case of only mixed NE is just as bad in the worst-case. (Consider an M x M rock-paper-scissors variant embedded within the NxN game. Then, global PSRO is likely similarly bad in the worst-case but more expensive than RGB-MSS variants.) A statement such as “incorporating appropriate global information into population expansion can dramatically improve iteration efficiency” is not surprising at all.

## Reproducibility and Code

The third issue is regarding reproducibility. Methods such as PSRO are particularly interesting when trying to solve very large zero-sum games. Even if I am convinced that the method is superior than previous ones, the implementation overhead is always substantial compared to previous methods. Therefore, providing an implementation is necessary in my opinion so that these incremental improvements may be adopted at all. The paper does not indicate that the authors plan to publish the code. (I understand that currently publishing it is problematic due to anonymity. However, one can still commit to do that within the paper already.)

## Minor comments

1. I find the naming of theoretical results as “theorems” confusing. A theorem is usually considered a main result.
- Theorem 4.1 is a commonly known property in finite normal-form games. This should be made clearer. For example, […]local stability […] which we summarize in this well known statement.
- Figure 3 is ambiguous. I think it shall demonstrate the connected regions that map to a specific best response. The games are not sufficiently explained (AlphaStar is not described anywhere and it is not as unambiguous as Blotto or other standard cases) and the exact are not explained either. As it is now, I think one should delete the plot and spend a few more lines on discussing what this is and why the connected regions are important.
2. I would argue to introduce the zero-sum case only in Section 2.1 as this simplifies the setting. This would clear up ambiguities regarding notation (U_i and U; n or N are the number of agents see beginning Section 2.1 but may also be the number of actions, see Equation 4).
3. There are some failed references to plots in the appendix (e.g., B.3.). Also, ensuring that the plots appear beneath corresponding headings (e.g., B.1 and B.2) would help the reader to find this easier.

---

> ### Author Rebuttal · Authors · 2026-03-31
>
> We thank the reviewer for the careful reading. We address the main concerns below.
>
> **1. Empirical cost accounting and fairness of comparison.**
>
> **Step accounting and fair comparison.**
> In all main plots, one step means one environment interaction step, and the x-axis reports the total number of environment interaction steps used by each algorithm. For a baseline PSRO method, this includes BR-training rollouts together with the payoff-estimation rollouts. For Global PSRO, this includes all rollout steps used for candidate generation in the exploration phase, BR training for PE estimation in the selection phase, and payoff estimation. Interaction with the environment is the core operation of these rollout-based methods, and in practice it is typically the main cost. Moreover, prior PSRO baselines are also evaluated in terms of environment interaction steps. Therefore, we compare methods by the total number of environment interaction steps.
>
> **RM steps.** The RM steps are internal regret-minimization updates performed during PE estimation in the selection phase and do not involve additional environment interaction, because they operate only on the rewards already collected from previous rollouts.
>
> **Computation budget.**
> When we say that costs are aligned across BR training, exploration, and PE estimation, we mean that, in each iteration, BR training in standard PSRO, BR training in Global PSRO's exploration phase and BR training in selection phase are allocated the same total number of environment interaction steps, use the same network architecture, and use the same number of gradient steps. This budget is fixed and does not scale linearly with $K$, i.e. the number of the candidates.
>
> **Additional computation for testing multiple candidates.**
> Apart from BR training, the additional cost of testing multiple candidates mainly comes from the RM (Exp3 in our experiments) updates and from computing PE estimates for the candidates. This overhead is slight in our experiments: RM uses only $100$ updates per candidate, each operating on a vector of size equal to the population size ($<100$ in our experiments), and PE estimates are computed by averaging rewards from rollouts already collected during RM-BR.
>
> **A concrete example.**
> In Leduc Poker, standard PSRO with RGB-MSS uses $8\times 10^5$ environment interaction steps to train one BR and then adds that strategy to the restricted population. In Global PSRO, the exploration phase uses a total of $8\times 10^5$ environment interaction steps to generate candidate strategies with a shared conditional policy, and the selection phase uses another $8\times 10^5$ environment interaction steps for PE estimation with a shared conditional evaluator. The selection phase also performs $100$ internal Exp3 updates for computing the least-exploitable mixture, which do not require additional environment interaction.
>
> **Restricted-game Nash in Table~1.** We use Restricted-game Nash as the base RGB-MSS in all games for consistency. We choose it as the concrete base MSS because it satisfies the convergence condition in Theorem 4.2.
>
> **2. Scope and insight of the theoretical results.**
>
> We agree that Theorems 3.2 and 3.3 are not general results for all zero-sum games. Our goal here is narrower: to use theory to isolate a concrete failure mode of restricted-game-based PSRO and, in turn, motivate algorithms that incorporate more global information. In this sense, Theorems 3.2 and 3.3 are meant as theoretically motivated evidence rather than a complete rate theory. Developing a broader and more systematic theoretical understanding remains important, and we view it as a natural direction for future work.
>
> **3. Reproducibility.**
>
> For reproducibility, we will release the code, configurations, and evaluation scripts in the revised version.
>
> **4. Minor comments.**
>
>
> **Theorem naming.** We agree that Theorem 4.1 is better presented as a proposition.
>
> **Figure 3.** We will add more description to clarify its meaning, and we will also consider moving it to the appendix if space is better used elsewhere.
>
> **Preliminaries / notation.** We will clean up the presentation and notation in the revised version.
>
> **Appendix references.** We will fix these issues in the revised version.

---

> > ### Author Rebuttal · Reviewer_DLt3 · 2026-04-02
> >
> > I thank the authors for addressing my concerns.
> >
> > The clarification of how "step" is counted precisely allows me to interpret the results better. I advice to include some of this explanation into an updated version of the paper. I agree that this is the appropriate way to present the results in the main body.
> >
> > I increased my score but I am still not conviced for publication.
> >
> > The use case of PSRO-type algorithms is to solve large zero-sum games. Whether the proposed method is beneficial for that purpose is not demonstrated. First, the theoretical results do not concern the hard case of mixed strategies, where worst-case should be just as bad as other methods. I understand that the purpose of the theoretical statements is to highlight a specific shortcoming of previous methods. However, the field is interested in solving hard cases and the presented theory provides no benefit to this. Second, the performed empirical experiments remain limited. After re-reading, I noticed that the largest game (Goofspiel) was only with five cards. Testing on larger instances (using approximately optimal solutions as baseline, see e.g., the results in Anytime-PSRO, McAleer et al. with Goofspiel with 13 cards) are necessary to test the method in games that are closer to relevant applications for the field.

---

> > > ### Author Response · Authors · 2026-04-08
> > >
> > > Thank you for the follow-up. We are glad that our explanation of the experimental setting addressed your concern, and we will revise the main paper to describe the experimental protocol in the same clear way.
> > >
> > > We agree that whether the method is beneficial for the intended PSRO use case of large zero-sum games requires further evidence. To provide such evidence, we added a larger-scale experiment on Goofspiel with 13 cards (Goofspiel-13). All methods are compared under the same total number of environment interaction steps, and each method is run independently four times. We use estimated PE as the performance metric, with PE estimated in the same way as in our Goofspiel experiments in the paper. The new results are shown in Table 1 (more detailed results and hyperparameter settings are provided in [ https://anonymous.4open.science/r/ICML2026-rebuttal-D13B/Tabs_for_Goofspiel13.pdf ]). On Goofspiel-13, Global PSRO achieves the lowest estimated PE at nearly all reported environment-step levels. We will revise the paper to make this larger-scale evidence more central in the experimental section.
> > >
> > > **Table 1.** Estimated PE at different environment steps on Goofspiel-13.
> > > **Environment steps (×10^6)**
> > >
> > > |Method|38.4|76.8|115.2|153.6|
> > > |---|---|---|---|---|
> > > |**Global PSRO (ours)**|**0.16 ± 0.04**|**0.06 ± 0.04**|**0.08 ± 0.07**|**0.05 ± 0.02**|
> > > |PSRO+Nash|0.41 ± 0.06|0.28 ± 0.04|0.21 ± 0.02|0.19 ± 0.04|
> > > |PSRO+PRD|0.57 ± 0.10|0.50 ± 0.06|0.45 ± 0.06|0.37 ± 0.01|
> > > |PSRO+Alpharank|0.29 ± 0.04|0.19 ± 0.00|0.16 ± 0.07|0.18 ± 0.05|
> > > |PSRO+Uniform|0.37 ± 0.00|0.28 ± 0.04|0.20 ± 0.04|0.23 ± 0.00|
> > > |Anytime PSRO|0.22 ± 0.08|0.22 ± 0.10|0.18 ± 0.06|0.19 ± 0.06|
> > > |NeuPL+Nash|0.30 ± 0.04|0.23 ± 0.02|0.16 ± 0.04|0.14 ± 0.04|
> > > |NeuPL+PRD|0.50 ± 0.06|0.44 ± 0.07|0.37 ± 0.03|0.33 ± 0.01|
> > > |NeuPL+Alpharank|0.21 ± 0.01|0.17 ± 0.01|0.19 ± 0.02|0.13 ± 0.04|
> > > |NeuPL+Uniform|0.30 ± 0.05|0.25 ± 0.04|0.21 ± 0.01|0.17 ± 0.00|
> > > |PSD-PSRO+Alpharank|0.35 ± 0.02|0.25 ± 0.00|0.22 ± 0.01|0.16 ± 0.01|
> > > |PSD-PSRO+Nash|0.49 ± 0.03|0.34 ± 0.06|0.28 ± 0.04|0.21 ± 0.03|
> > > |PSD-PSRO+PRD|0.68 ± 0.08|0.56 ± 0.06|0.44 ± 0.01|0.31 ± 0.00|
> > > |PSD-PSRO+Uniform|0.34 ± 0.00|0.28 ± 0.03|0.24 ± 0.02|0.15 ± 0.02|
> > >
> > > ***
> > > On the theory side, our analysis shows that, in worst-case scenarios, PSRO with RGB-MSS may need to add all pure strategies to reach equilibrium, and for some RGB-MSSs, may even fail to converge. By contrast, Global PSRO retains a convergence guarantee (Theorem 4.2) and can be faster in some worst-case scenarios (Theorem 4.3). While this already reveals a concrete advantage in certain hard cases, we also agree that the current theoretical analysis is limited to games with pure-strategy NE.
> > >
> > > However, the theoretical advantage in Theorem 4.3 can be extended to the mixed-strategy setting. We provide a proof sketch below. The extension builds on the same symmetric zero-sum game constructed in Theorem 4.3, but replaces the pure-strategy Nash equilibrium with a mixed-strategy Nash equilibrium supported on a strategy subset $T$. We then construct the payoffs involving strategies in $T$ in the full game to ensure faster convergence of Global PSRO. The guiding principle is simple: strategies in $T$ should have an advantage against strategies outside $T$, and the payoffs among strategies in $T$ should remain small.
> > >
> > > The payoff construction is designed to ensure the following requirements:
> > >
> > > (1) The full game admits a Nash equilibrium whose support is exactly $T$.
> > >
> > > (2) For PSRO with RGB-MSS, the same worst-case behavior is preserved: it first adds the strategies outside $T$, and then must continue adding all strategies in $T$ in order to reach equilibrium.
> > >
> > > (3) For Global PSRO, after one iteration (corresponding to two PSRO-style iterations), every strategy in $T$ can already be induced by some meta-strategy over the current restricted game (analogous to the pure NE in Theorem 4.3).
> > >
> > > (4) After the iteration, whenever Global PSRO has not yet reached a Nash equilibrium, any best response induced by the current restricted game that minimizes PE must lie in $T$.
> > >
> > > Therefore, after the iteration, Global PSRO only needs to identify suitable strategies from $T$ among its candidates, rather than traverse the entire strategy set. This leads to an upper bound on the expected number of PSRO-style iterations:
> > >
> > > $$\min\left(2+\frac{2|T|}{1-(1-c^\prime)^{K-1}},N-1\right),$$
> > >
> > > where $c^\prime\in(0,1]$ is a game-dependent constant, $K$ is the exploration-pool size, and $N$ is the number of strategies in the full game. In particular, when $K$ is sufficiently large, the bound approaches $\min(2+2|T|,N-1).$
> > > The above requirements induce a game with a mixed NE in which PSRO with RGB-MSS still exhibits worst-case behavior, while Global PSRO reaches equilibrium faster. However, the construction details and the proofs of the above properties are lengthy, and due to the rebuttal space limit, we cannot include them here; we promise to add them in the revised version.

---

### Official Review · Reviewer_sNb5 · 2026-03-12

**Soundness:** 3
**Presentation:** 3
**Significance:** 2
**Originality:** 2
**Overall Recommendation:** 3
**Confidence:** 4

**Summary:**

The paper proposes Global PSRO, a novel framework that addresses the challenge of inefficient population expansion in Policy-Space Response Oracles (PSRO) by directly optimizing the quality of the expanded strategy set. It introduces a two-phase exploration–selection mechanism that minimizes Population Exploitability (PE)—a measure of how well the restricted set approximates the full game—to guide the expansion process.

**Compliance With Llm Reviewing Policy:**

Affirmed.

**Key Questions For Authors:**

1. In the experimental results, for Kuhn Poker and Liar's Dice, multiple algorithms showed a significant decrease in PE after a certain number of steps. However, this phenomenon was not observed in the other two games. Why is that?
2. Regarding the experiment, the benchmark method PSRO should be considered. Considering the small scale of the evaluated games, it is worth considering whether the classic CFR series algorithms of solving such games have been taken as the baseline for evaluation.
3. The paper considers the global population, and how the number of individuals in the population is set. Since the proposed method expands from the global perspective, how does it compare with the basic algorithms in terms of memory, duration, etc.? Providing the codes of all algorithms in the experiment may better evaluate this work.

**Limitations:**

yes

**Strengths And Weaknesses:**

Strengths: The organization of the paper is very good.
Weaknesses: This is an incremental innovation.  The objectivity and reproducibility of the experimental results still need to be verified.

---

> ### Author Rebuttal · Authors · 2026-03-31
>
> Thank you for the careful review. We address the main concerns below.
>
> **“For Kuhn Poker and Liar’s Dice, PE drops sharply after some steps, but not in the other two games. Why?”**
>
> In Kuhn Poker and Liar’s Dice, which are relatively smaller games, the equilibrium can be supported by relatively few key pure strategies. Once these strategies are included in the restricted population, PE drops sharply. In Leduc Poker and Goofspiel, the game is relatively larger, so collecting the necessary support strategies is harder and the decrease in PE is therefore more gradual.
>
> **“Regarding the experiment, the benchmark method PSRO should be considered. Considering the small scale of the evaluated games, should CFR-style algorithms also be used as baselines?”**
>
> We have already compared against a broad set of PSRO baselines in our experiments. We agree that classical CFR methods are important references for small extensive-form games. However, CFR and PSRO operate in different settings: CFR assumes an explicit extensive-form model of the full game and performs model-based updates on that structure, whereas PSRO is model-free and instead relies on simulator rollouts, learned best responses, and iterative population expansion. For this reason, CFR-style methods are not the apples-to-apples baselines for the question studied here.
>
> **“How does it compare with the basic algorithms in terms of memory, duration, etc.? Providing the codes ... may better evaluate this work.”**
>
> The dominant memory cost of both Global PSRO and standard PSRO comes from storing the policy networks, which scales linearly with the population size. Under the same environment interaction steps, the two methods construct strategy populations of the same size, so their memory usage is essentially the same.
>
> The runtime of both Global PSRO and standard PSRO is dominated by neural-network training and payoff-matrix evaluation. Under the same population size, these costs are essentially the same for the two methods. The additional runtime of Global PSRO comes from the selection stage, which introduces extra computation for PE estimation via regret minimization. In practice, this overhead is small. On our hardware platform (Intel Xeon Gold 6348 CPU @ 2.60GHz, 8 NVIDIA RTX A6000 GPUs, and 512 GB memory, running Ubuntu 18.04), with no other workloads running, we independently ran each method four times. In Leduc Poker at 40M environment interaction steps, standard PSRO takes $9.04 \pm 0.38$ h, while Global PSRO takes $9.38 \pm 0.29$ h, which is about $4$% more.

---

### Decision · Program_Chairs · 2026-04-30

**Decision:**

Accept (regular)

**Comment:**

This paper studies policy-space response oracles (PSRO) for two-player zero-sum games and proposes a modified expansion mechanism along with a new evaluation perspective based on population exploitability to guide policy population growth.
Reviewers generally agree that the paper is well-structured, technically sound, and addresses an important problem in scalable equilibrium computation, with reasonable empirical evidence across several benchmarks. In particular, the introduction of a two-phase selection/expansion process and the focus on minimizing population exploitability are viewed as meaningful extensions of prior PSRO frameworks. While some concerns were raised regarding the clarity of the theoretical justification, fairness of certain experimental comparisons, and the degree of novelty, these issues were partially addressed during the rebuttal and discussion. I therefore recommend a weak accept.